# Reconstructing missing complex networks against adversarial interventions

Yuankun Xue (iD) [1] & Paul Bogdan (iD) [1]

Interactions within complex network components define their operational modes, collective behaviors and global functionality. Understanding the role of these interactions is limited by either sensing methodologies or intentional adversarial efforts that sabotage the network structure. To overcome the partial observability and infer with good fidelity the unobserved network structures (latent subnetworks that are not random samples of the full network), we propose a general causal inference framework for reconstructing network structures under unknown adversarial interventions. We explore its applicability in both biological and social systems to recover the latent structures of human protein complex interactions and brain connectomes, as well as to infer the camouflaged social network structure in a simulated removal process. The demonstrated effectiveness establishes its good potential for capturing hidden information in much broader research domains.

[1] Ming Hsieh Department of Electrical and Computer Engineering, University of Southern California, Los Angeles, CA 90007, USA. Correspondence and requests for materials should be addressed to P.B. (email: pbogdan@usc.edu)

The interaction structure largely determines the operation modes, collective behaviors, and global functionality of complex systems. Consequently, it is crucial to discover the best identification and recovery strategies for sabotaged networks subject to unknown structural interventions or camouflages. Due to its broad applicability, this problem draws interdisciplinary attention in research areas ranging from network science[1–4], social science[5], system engineering[6,7] to ecology[8], systems biology[9], network medicine[10], neuroscience[11,12], and network security[13] communities. This challenge raises intense research interest in recent years[14–23]. By contrast, very few works discuss and incorporate the statistical influence of the interventions. Most prior works assume that the structural intervention entails a sequence of randomly distributed removals of nodes and links in the network. Under such assumption, constructing an unbiased estimator for the nodal or edge property (but not both at the same time) is shown to be possible[14] and can be approached by solving a matrix completion problem (e.g., low-rank matrix factorization[15–17], convex optimization[18,19], spectral methods[20]). These methods become mathematically infeasible when nodes and edges are simultaneously removed. Extending existing approaches to deal with such problems requires additional information that links the known and unknown part of the network (e.g., group membership[21,22] and node similarity[23]) and become obsolete when such information is not available. Alternatively, model-based approaches are adopted in these settings by learning a probabilistic connection between the observed and the latent network structure. These probabilistic links are parametrized and identified in a maximum likelihood sense.

Many of these approaches[24,25] can be unified within an Expectation-Maximization (EM) framework that solves the model identification and inference problems simultaneously through an iterative trial-and-error approach with a provable convergence to the local maxima of the incomplete likelihood function. However, in the context of the missing network inference subject to artificially (not randomly) introduced interventions, the latent structure does not share the identical distribution as the observed one, but follows a reshaped distribution. This invalidates the use of EM formulations based on the assumption of random network removals, which do not change the underlying distribution.

To overcome this challenge, we propose a causal statistical inference framework (see Methods). In contrast to prior efforts, our framework jointly encodes the influence of probabilistic correlation between the visible and invisible part of the network (i.e., network model) and the stochastic behavior of the intervention. More importantly, this inference framework captures the temporal causality of sequenced attacks and treats the partially observed network as a result of time inhomogeneous Markovian transitions driven by the intervention. The proposed inference framework can be applied with any underlying network models that are appropriate to the specific problem settings. As a case study, we employ the multi-fractal network generative (MFNG) model (see Supplementary Note 4) as the underlying network model[26] because it can model a variety of network types with prescribed statistical properties (e.g., degree distribution). To validate our framework, we discuss and evaluate it on both synthetic and real networks in biological and social domains.

## Results

**Motivating example and problem formulation**. For the success of an iterated inference within an EM framework, a combined modeling of the network and interventional behavior is necessary. Let us consider a toy problem in Fig. 1. An attacker removes node A from $G_0$. We observe $G_1$ as a resulting network after the attack. The problem is to infer $G_0$ from $G_1$. We make three assumptions: (i) The attacker always targets the most connected node. When such a node is not unique, it randomly chooses one. (ii) There is an underlying generative model for $G_0$ that discourages nodes of high connectivity and does not allow for disconnected nodes. (iii) We have perfect knowledge of both the attacker and the generative model.

According to the Bayesian inference principle, we infer the missing node and its links that maximize the likelihood based on the network model and the attacker's statistical behavior. By assumption (ii), the missing node inferred based on the network model will be less likely to have a higher degree. $G_{0,1'}$ therefore can be one of the possible outcomes ($G_{0,2'}$ represents another possibility). Although $G_{0,1'}$ is not unique, one must choose it over many other possible configurations where node A has a higher degree. By assumption (i), the missing network inferred based on the attack can be $G_{0',1}$, $G_{0',2}$, or $G_{0',3}$ (other outcomes removed due to symmetry). However, node A is not the unique most connected node in $G_{0',1}$ and $G_{0',2}$ (i.e., only 50% chance to be chosen). Therefore, $G_{0',3}$ is the most probable outcome. Interestingly, neither $G_{0,1'}$ nor $G_{0',3}$ represents the true configuration. From the perspective of the network model, $G_{0',3}$ is a less likely structure due to the highly connected node. $G_{0,1'}$ is less likely (1/3 chance) to be the target of the attacker. Combining the knowledge of both leads us to the true $G_0$ in this simple case.

Inspired by this example, we incorporate the attack model and formulate the challenge as a causal inference problem of time-varying complex networks under adversarial interventions as follows.

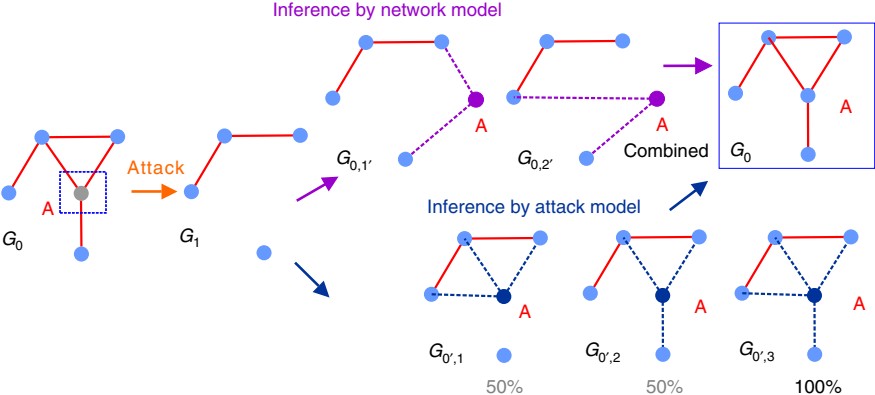

**Fig. 1** A motivating example. An illustrative example to show the importance of combined consideration of network model and interventional behavior

Given a partially observed network $G_t = (V_t, E_t)$. $G_t$ is a subgraph of an unknown network $G_0$ under structural intervention $\mathcal{A} = \{\mathcal{A}_\alpha(d_i, s)|s \geq 0\}$. Denote $\mathcal{G}$ to be a proper underlying model that captures the network properties such that $G_0$ is a realization of $\mathcal{G}$ and we denote it as $G_0 \models \mathcal{G}$.

Find the missing sub-network $M_t$ where $M_t \cup G_t = G_0$ and node-to-time mapping $\pi$ such that,

$$\text{argmax}_{\mathcal{G}, M_t, \pi} P(G_t, M_t, \pi|\mathcal{G}, \mathcal{A}) \quad (1)$$

The consideration of mapping $\pi$ in Eq. (1) comes from the causal interdependency on the transitional path from $G_0$ to $G_t$ due to the time varying interventional preference $\mathcal{A}_\alpha(d_i, s)$ being a function of $G_s$. In other words, a most probable sequence of interventions needs to be discovered so as to maximize the posterior in Eq. (1). As a result, any missing substructure in $M_t$ has to be placed properly in time subject to the causality, hence the requirement of the node-to-time mapping $\pi$. The statistical strategy of the intervention is characterized by a power-law family of distributions[27],

$$\mathcal{A}_\alpha(d_i, t) = \frac{d_i^\alpha}{\sum_i^{N(t)} d_i^\alpha} \quad (2)$$

where $\mathcal{A}_\alpha(d_i, t)$ denotes the probability of a node $i$ of degree $d_i$ to be removed from a time-varying network $G_t = (V_t, E_t)$ at time $t$. $N(t)$ is the total number of nodes at time $t$. $\alpha$ is a parameter that governs the statistical property of the adversarial intervention distribution. When $\alpha > 0$, the intervention prioritizes high degree nodes (hubs). Such interventions are observed in real systems obeying small-world principle[28]. Small-world networks are known to be robust against random removals, but vulnerable to hub-prioritized attacks. For example, in biological systems, viral attackers have evolved to exploit the small-world properties and interfere in the hub proteins activity such as p53, thereby taking advantage of cellular functions for fast viral replication[29]. In contrast, when $\alpha < 0$, the intervention strategy focuses on less connected nodes (i.e., boundary nodes). For instance, in computer networks, boundary nodes usually correspond to end-users with less security measures to protect their devices, thereby becoming prey to malicious hackers and malware. Random attacks are performed when $\alpha = 0$ and all nodes have an identical chance to be removed. More critically, Eq. (2) is a function of $G_t$ hence a time-varying distribution, which suggests the causal dependency of interventions. At a given time $s$, $G_s$ is a causal consequence of all intervention sequences prior to that time point. From a dynamic perspective, this time-varying distribution of the intervention leads to a time-inhomogeneous Markovian transition of $G_t$ between different configurations in time. To better understand this aspect, Supplementary Note 1 provides a detailed discussion of an example concerning this inference problem.

To solve the problem in Eq. (1), we propose a causal statistical inference framework (see Methods). To evaluate the framework, we consider two set of experiments with synthetic and real networks. In both experiments, we assume MFNG model $\mathcal{G}_k = (m, k, \mathcal{P}, \mathcal{L})$ (see Supplementary Note 4 for detailed discussion) and compare our proposed framework against a baseline where the discount factor $\gamma$ is fixed to 0 to ignore the influence of the attack.

**Reconstruction against ground-truth**. We first test how well the inference framework can retrieve the original network if we have perfect knowledge about the model that generates it. We sample synthetically a test network $G_0$ of 1024 nodes ($k = 10$, $m = 2$) with a randomized generating measure $\mathcal{P}$. Then the intervention $\mathcal{A}_\alpha(d_i, t)$ is introduced sequentially for $T$ steps,

where $T$ ranges from 5% to 45% of the total number of nodes in the original network. We also vary the statistical preference of the intervention by setting $\alpha$ differently to be 10 (hub-prioritized attack) and $-10$ (boundary-node prioritized attack). These values correspond to two distinct attack strategies that also influence the network inference process (as discussed later). Each intervention process is repeated for 10 times for every combination pair of $(T, \alpha)$. Denote the estimated generating measure induced by $\mathcal{G}$ as $\hat{\mathcal{P}}$ and the true one as $\mathcal{P}^*$. We report first the estimation error as the Frobenius norm $e_F$ of their difference to quantify the capability to recover the generating measure $\mathcal{P}$. Figure 2 shows the results averaged over 10 intervention trials as a function of amount of missing information. In contrast to the baseline, the estimation error of the proposed method is robust against the loss of network structural information and delivers accurate estimation of the underlying parameters even when 45% of the network is structurally sabotaged by the intervention. More importantly, the estimation error for the baseline is significantly larger than the proposed approach even for small percentage of network information loss (5–10%).

These results demonstrate the importance of accounting for the effect of intervention on the network probability measure. EM-type inference methods essentially construct the maximum likelihood estimator based on iteratively optimized incomplete likelihood function (i.e., Q-function). Instead of solving analytically this Q-function, Monte Carlo method highly relies on being able to draw samples of the latent variables (e.g., the missing network) from a distribution that is increasingly approaching their true distribution. As a result, the estimator converges to local maxima in the statistical manifold (as the generating measure $\mathcal{P}$ uniquely defines a distribution on a unit square). However, if the samples of the latent variables are always drawn from a distribution that is significantly different from the true distribution, it is unlikely that the estimates will be close to the true parameters and the resulting deviation increases with higher dimension of latent space (e.g., number of missing nodes increases).

Unfortunately, this is exactly how the baseline fails. The network model and the interventions now jointly determine the distribution of missing network. For instance, the degree distribution of victim nodes under a hub-prioritized intervention must concentrate the probability mass to the regions of relatively high degree (right-shifted in relation to what network model suggests). Failure to draw samples of the latent variable from their true distribution leads to large errors in model estimation (Fig. 2a). This will eventually affect the inference accuracy. To see this, we visualize the degree distribution of missing nodes and that supported by the true underlying model in Fig. 3a via kernel smoothing method. 40% of nodes and their links were removed with $\alpha$ ranging from $-10$ to 10. As predicted, the degree distribution of missing network concentrates increasingly its mass to the region of high degree as $\alpha$ becomes positively larger. Similar observation is due when $\alpha$ becomes negatively smaller. In either case, they are significantly shifted from the degree distribution supported by the network model (blue bold line), which explains the large estimation error of the baseline. More precisely, Fig. 2c, d report the Kullback–Leibler (KL) divergence $e_{KL}$ as a function of $\alpha$ and amount of lost information. Figure 2c shows that the baseline always underestimates (i.e., positive KL divergence) the linking probability of the missing nodes when $\alpha = 10$ and overestimates (i.e., negative KL divergence) it in Fig. 2d when $\alpha = -10$. This shows that the baseline neglects the intervention influence and suffers from large estimation errors.

To better illustrate this, Fig. 3b, c shows two degree distributions of the missing network recovered by the baseline

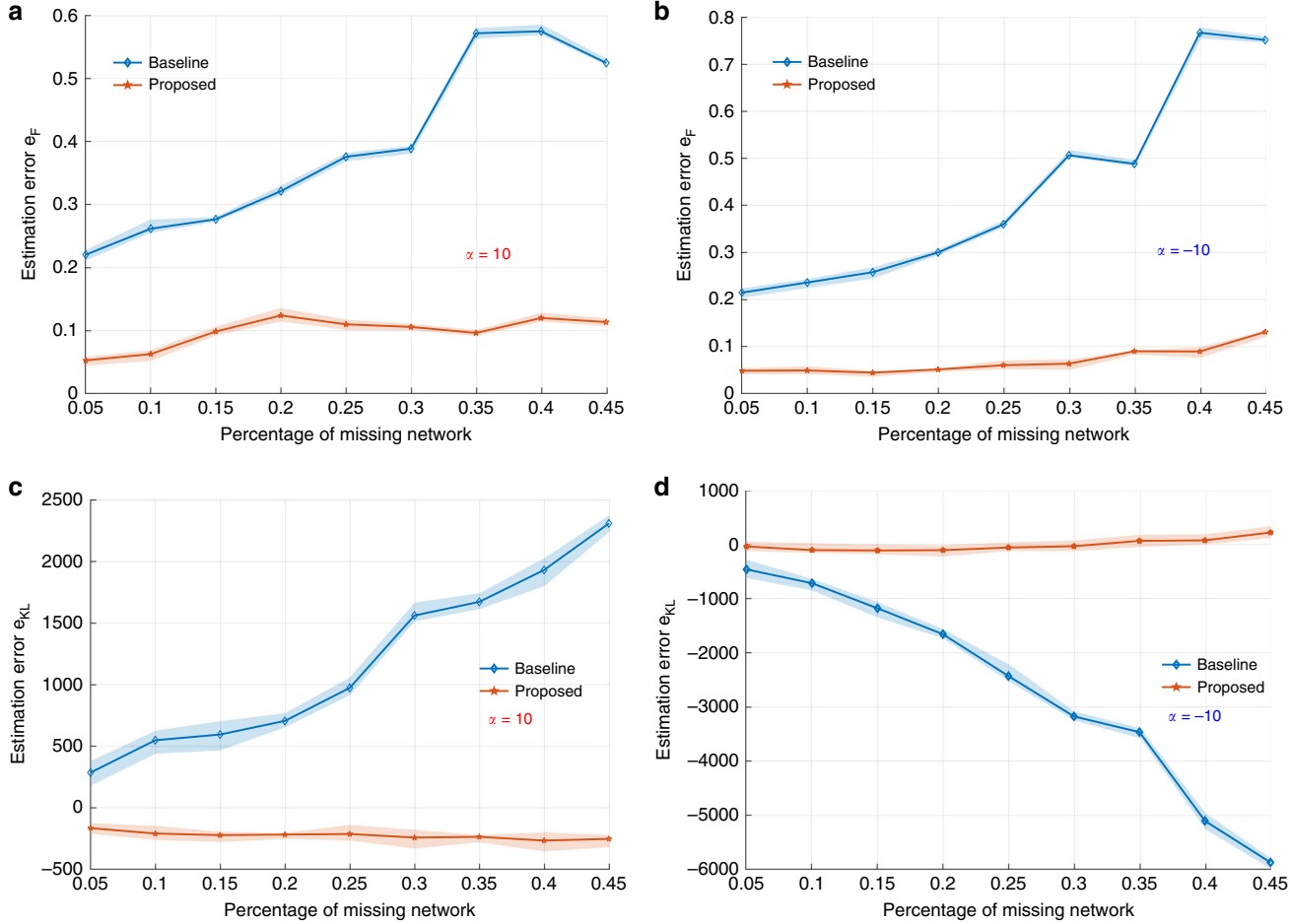

**Fig. 2** Quantifying the capability of inferring synthetic networks under varying attack strategies. **a**, **b** Estimation error $e_F$ as a function of missing network under hub-prioritized (**a**) and boundary-prioritized (**b**) intervention ($\alpha = 10$ or $-10$). **c**, **d** KL divergence comparison of the true linking probability distribution and the recovered ones by baseline and proposed frameworks with the presence of hub-prioritized (**c**)/boundary-prioritized (**d**) intervention ($\alpha = 10$ or $-10$)

and proposed methods. Figure 3b shows that the degree distribution retrieved by the baseline shifts greatly to the left of the true one (underestimation) when $\alpha = 10$ and the situation is reversed (overestimation) when $\alpha = -10$. In contrast, the proposed method recovers the distribution well in both cases. Our method incorporates the influence of the attack on the inference and takes only samples (as in the Monte Carlo process) approved by both the model and the attacker. Consequently, it is robust against the loss of information and delivers accurate estimations.

**Uncover the latent structure of biological systems.** In the first set of experiments with real networks, we demonstrate that the proposed framework recovers the latent gene interaction and brain networks when exposed to simulated targeted attacks. Attackers like virus or cancer cells in these systems usually do not possess the knowledge of the full network. However, the rationale for considering targeted attacks on these systems is that, when global information is not available, the probability of reaching a particular vertex by following a randomly chosen edge in a graph is proportional to the vertex's degree[30]. This makes the degree centrality an important factor in quantifying the vulnerability of the nodes, even if the attacker has only extremely localized information (e.g., connectivity). This resonates well with some of our biological findings in terms of viral spreading[31,32] and protein inhibition[33].

We consider a targeted attack process in two biological networks (hu.MAP and human brain connectome) with $\alpha = 1$ that models the hub-preferential interventions observed in real systems. hu.MAP network[34] encodes the interactions of human protein complexes. hu.MAP is a synthesis of over 9000 published mass spectrometry experiments containing more than 4600 protein complexes and their interactions. Of all protein complexes, we have identified the largest connected component consisting of 4035 protein complexes and used it as our target network. Budapest Reference Connectome v3.0 generates the common edges of the connectomes of 1015 vertices. It is computed from the MRI of the 477 subjects of the Human Connectome Project 500-subject release[35]. We vary the percentage of missing network nodes from 5% to 45% under a simulated attack that removes nodes. Both ROC-AUC and PR-AUC scores are computed under varying range of thresholds to quantify the inference capability of the models retrieved by baseline and our framework.

For hu.MAP network, Fig. 4a shows that the ROC-AUC score stays around 0.88 with only a small decrease to 0.85 when 45% of nodes are removed. In contrast, the ROC-AUC score of the model retrieved by the baseline degrades sharply from 0.85 to 0.68. Similar observations are due for the PR-AUC score where proposed framework raises it from 0.17 to 0.23 with 5% of node loss and from 0.15 to 0.21 when 45% of nodes are removed. We note that the PR-AUC score is much lower as compared to ROC-AUC. This is due to sparsity of the network. The number of links

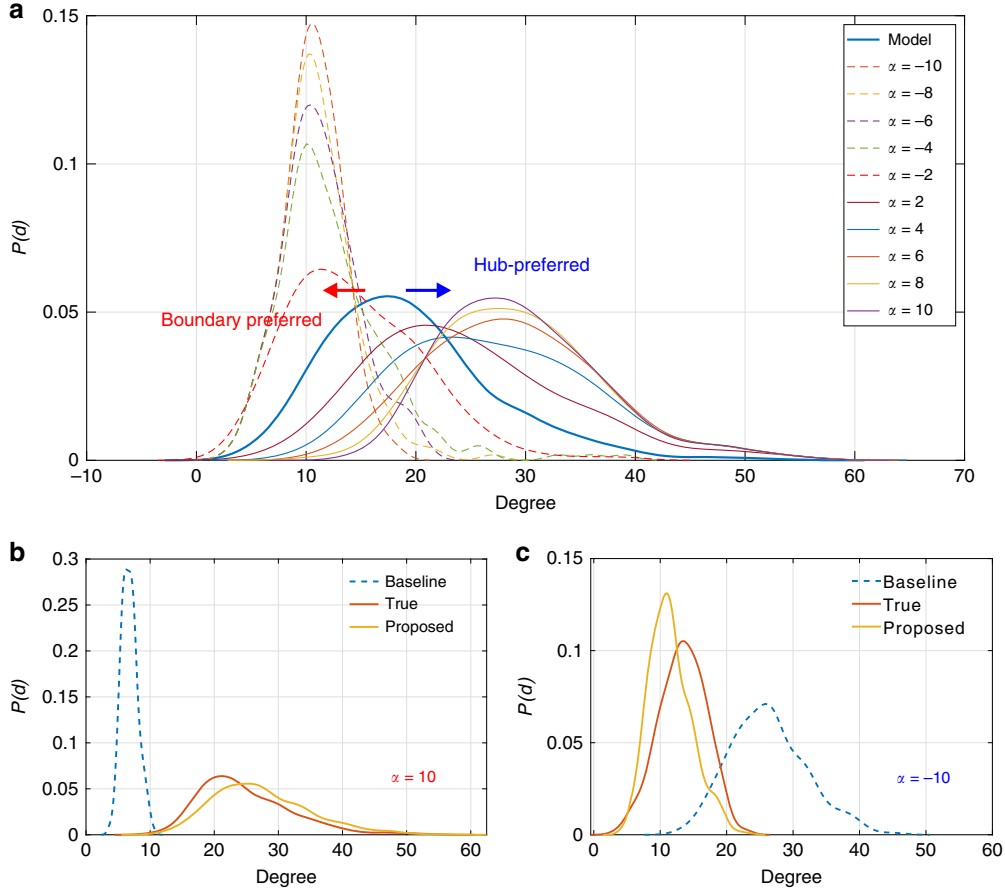

**Fig. 3** Degree distribution analysis. **a** Reshaped degree distribution of the latent network structure. **b**, **c** Capability to reconstruct the missing network structure under hub-prioritized intervention ($\alpha = 10$, **b**) or boundary-prioritized intervention ($\alpha = -10$, **c**) with 45% of nodes removed

(i.e., positives) is much smaller than that of a complete network of the same size and both methods produce noticeable amount of false positives. The source of these false positives can be (i) insufficient order of the model (e.g., choose larger $k$ for linking measure matrix, see Supplementary Note 4), (ii) insufficient sample size in E-step, (iii) overshooting in M-step. While realizing the space for fine-tuning and improvement, we note that the proposed framework places no constraint on the proper choice of model and its real power lies in considering and exploiting the influence of interventions, rather than treating them as a random sampling process.

For human brain connectome, we observe a slightly different pattern in Fig. 4e. While ROC-AUC score obtained by the proposed method is consistently higher than the baseline, the score of both methods degrade first (up to 15% of nodes removed) and then oscillate afterwards. This phenomenon is due to three facts: (i) human brain connectome is rich in small-worldness; (ii) there are much fewer hubs in brain connectome than in hu.MAP; (iii) the intervention becomes close to a random sampling after most of the hubs are removed and small-world networks are robust against such random removals. As a result, the attack process quickly reduces to a random sampling after the few hubs are removed. Thereafter, the residual network loses the structural resemblance to the original network, which serves as the very basis for EM-type inference methods to work. Averaging out the contribution of latent structure in the E-step now effectively wipes out the structural properties of the original network to be recovered (as it becomes dominant now). This leads the iterative optimization process of EM to a nondeterministic search in the solution space (which is super-exponentially

large), leading to predictions that are not aligned with the original networks. However, even under such conditions, our framework consistently recovers the network that is more structurally similar to the original one. This resonates again with our argument in hu. MAP experiment that exploiting the combined knowledge of the generative model and the intervention can significantly boost the performance. Similar observations are due for PR-AUC scores.

To quantify the capability to recover the global property of the original network, we use the log-likelihood and KS distance. The KS distance is averaged over 1000 network samples drawn from both models and shown in Fig. 4b, f, respectively. The solid lines in both figures represent the averaged distance with the shades being the standard deviation. In both figures, the KS distance of the generated network via our proposed method is consistently robust to the interventions and more accurately retrieved than the one obtained by the baseline approach, which is an indicator of a boosted structural similarity between the true one and the synthesized ones. To further support our findings, we compute the log-likelihood function in Fig. 4d, h, respectively, based on both models with respect to the original network.

We notice that the two figures are similar to each other, suggesting that the overall goodness-of-fit of the identified model highly relies on being able to guide the optimization in EM framework iteratively towards a linking probability measure (i.e., a network model) that best explains the original network. Otherwise, the error can easily propagate repetitively between the inference and the estimation step, resulting in a retrieved model that poorly explains the original network as we have seen in these two figures. Both methods perform similarly to fit a model that explains the observed part of the network. However, the baseline retrieves

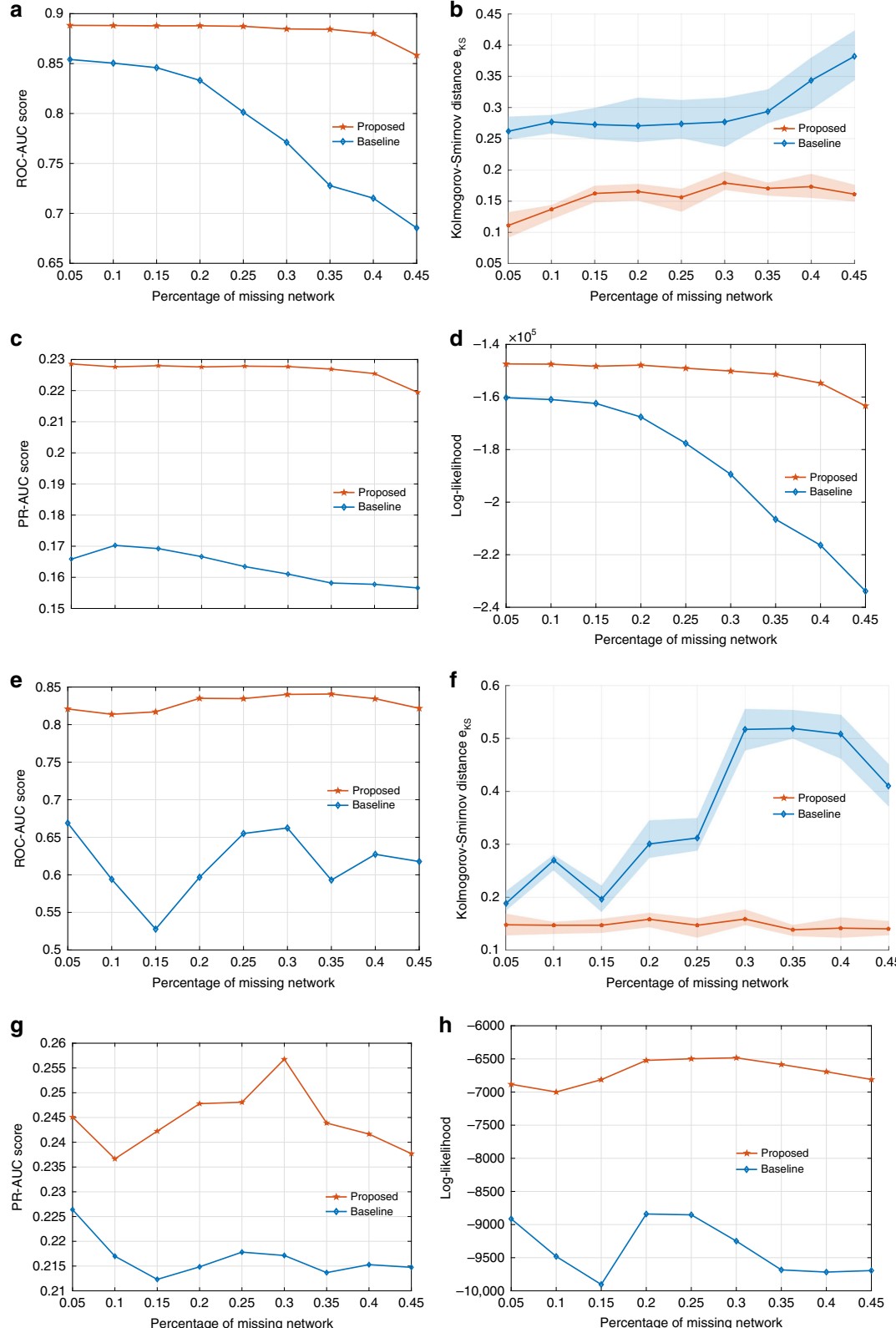

**Fig. 4** Recover the human protein complex interaction network (**a**–**d**) and brain consensus connectome (**e**–**h**). **a**, **c**, **e**, **g** Compare the capability to infer the missing network via AUC score. **b**, **f** Goodness-of-the-fit comparison as reported by Kolmogorov–Smirnov distance between the true degree distribution and the one retrieved by baseline and proposed methods. **d**, **h** Quantification of the capability of both methods to recover the global property of the protein complex interaction via log-likelihood

models incapable of inferring the latent structure as accurately (quantified by the AUC scores) as our proposed method does. Consequently, the difference of log-likelihood in terms of the latent structure dominates, hence producing a similar pattern between AUC score curve and log-likelihood curve.

In summary, we observed a significant boost in structural similarity by our framework that incorporates and exploits the influence of the interventions on the underlying distribution of the latent structures of the two studied biological networks as compared to the baseline that treats the unobserved and observed networks in a statistically equal way (i.e., random sampling assumption).

**Discover the hidden social networks**. Next, we use our framework to discover the hidden subnetwork in a simulated removal process that mimics the social network interventions in an abstracted setting. This study is inspired by the recent social network user privacy and information breaches via injected malicious agents (trolls and bots)[36,37]. These injected agents act as information collectors or launch campaigns to propagate designed information to target social groups. Together with the user nodes, they form an extended network that is usually not fully unveiled. The ultimate challenge is to estimate their structural formation and influence on various social events. Although real social network attacks can be much more sophisticated by involving multiple parties at the same time (as opposed to a coordinated sequence of operations as in Eq. (2)), evolving in a statistically inconsistent way (as opposed to a stabilized and consistent stochastic behavior) and exhibiting a complex opinion diffusion dynamics, we here consider an idealized abstraction of a class of real attacks that prioritize the degree centrality. The considered attack model and its variants have been widely adopted as an abstraction of the targeted attacks for the study of robustness, stability, resilience, and defensive/attack strategies of networks[27,30,38–44] ranging from mathematically constructed complex network to traffic network[45], brain network[46–48], computer network[13], and also social networks[49,50].

We consider an extended social network with 4049 nodes (including hidden nodes injected for information manipulation, referred as injected nodes, and ordinary user nodes) built from Facebook network dataset[51]. Due to the small-worldness of the social network (see Supplementary Note 2), only a small group of injected nodes is required to make sure all user nodes have at least one injected node as their immediate neighbor (i.e., all users are subject to data security issues and/or manipulated information even without information propagation among them). We define coverage to be the chance of a user node to have an immediate neighboring injected node. Figure 5 visualizes the coverage of injected nodes against their share in the network under different $\alpha$ from $-10$ to $10$. In this figure, $\alpha$ has a different meaning and $\mathcal{A}_\alpha(d_i)$ now is a proxy of the likelihood of an injected node of degree $d_i$ being the highest connected node in the network. For higher $\alpha$, a larger portion of the highest connected nodes are represented by injected nodes and so they have a bigger coverage. Figure 5 suggests that 48.6% of the population have at least one neighboring injected node when the injected nodes account for only 1% of total nodes with $\alpha = 1$. The coverage goes up to 98.44% when injected nodes account for 15% of the network as shown in Fig. 5. This suggests that a full-scale information manipulation/collection requires only a small injection of designed agents (i.e., disseminators/collectors) into the network and these agents do not have to be significantly more connected than an average node. This observation is corroborated by a recent study of Russian trolls attack on Twitter[52] which found that the injected tweet bots only account for 4.9% and 6.2% of total liberal and conservative spreaders, respectively.

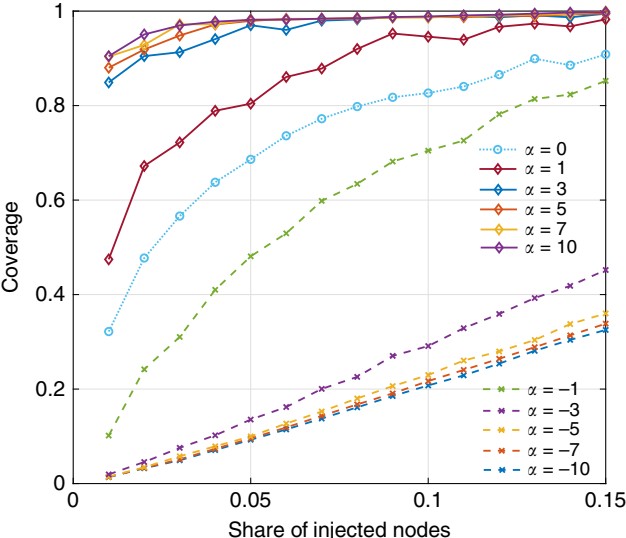

**Fig. 5** Coverage as a function of $\alpha$ and share of injected nodes in total network of 4049 nodes

We simulate the removal process by setting $\alpha = 1$ and vary the share of injected nodes from 5% to 45%. ROC-AUC and PR-AUC are used as metrics for quantifying the inference capability and shown in Fig. 6a, c for baseline and proposed methods. Resonating with our previous experiments, the ROC-AUC and PR-AUC scores of our proposed inference framework are significantly better than the baseline, suggesting a boost in capability to infer the missing network more accurately. To measure the structural similarity, we estimate the Kolmogorov–Smirnov (KS) distance $e_{KS}$ between the empirical degree distribution of the original network $F^*(x)$ and networks generated by both methods $F(x)$. The results are averaged over 1000 network instances and reported in Fig. 6c. In addition, we also report the log-likelihood (LL) in Fig. 6d as a global metric for goodness-of-fit to compare the model identified by both methods. Although the absolute value of LL strongly varies as a function of a particular model choice for the network, the relative difference given the fixed model provides a good performance comparison between different identification techniques. As expected, Fig. 6b, d suggests that our proposed method retrieves a model that is more globally consistent with the true one with smaller $e_{KS}$ and larger LL values compared to the baseline.

The statistics of both the intervention process and the complex network structure play a crucial role in these observations. First, in small-world networks, the hubs account for a small fraction of the network. Lower degree nodes are unaffected by hub-prioritized interventions. The baseline ignores the influence of the intervention and therefore is biased by the observed part towards the retrieval of a model that explains better a network without the hubs. As demonstrated by our studies, the baseline has poor performance on inferring the missing network. Second, due to the time-varying nature of the interventions, the hub-prioritized interventions induce a random sampling behavior after the removal of hubs. This behavior change can be demonstrated by the small variance of the degree distribution, reshaped by the conducted intervention (see Transitional behavior of interventions in Supplementary Note 3). Consequently, the performance of baseline and proposed methods exhibit a plateau since a small-world network is robust against random removals. We present the investigation of small-world-ness of all networks considered in our work in Supplementary Note 2.

Last but not least, we report the estimated number of user nodes (later referred as "affected users") with at least one injected

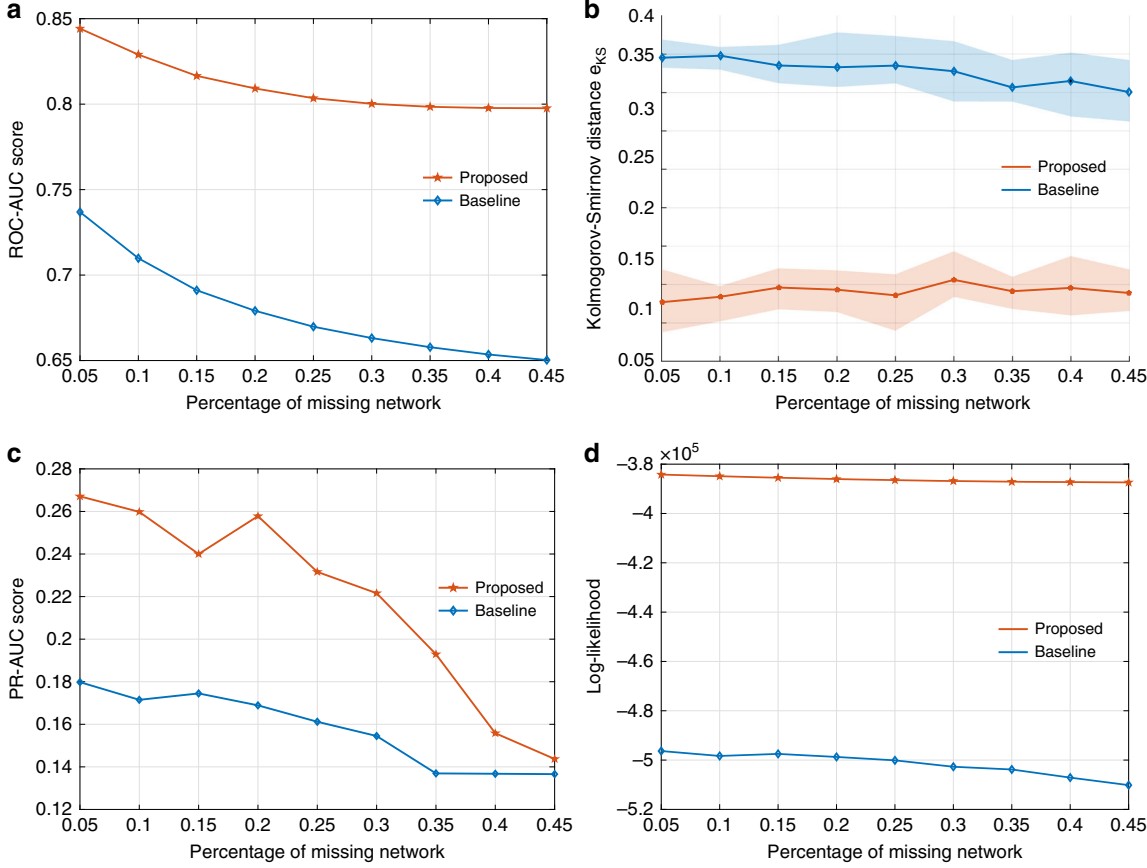

**Fig. 6** Evaluation of the capability to recover the Facebook social network ($\alpha = 1$). **a**, **c** Compare the capability to infer the missing network via AUC scores. **b** Goodness-of-the-fit comparison as reported by Kolmogorov–Smirnov distance between the true degree distribution and the one retrieved by baseline and proposed methods. **d** Quantification of the capability of both methods to recover the global property of the example social network via log-likelihood

node as their immediate neighbor. Without considering the opinion diffusion dynamics, this measurement serves as an upper bound on the number of users being exposed to designed information or personal data breaches. To consider a more realistic setting, this assessment should also incorporate the propagation of information among users, which is left as an important extension in our future work. Varying the share of injected nodes in the extended social network from 1% to 15%, Fig. 7 shows the average affected users estimated over 5000 network instances drawn from both models retrieved through baseline and proposed methods. As expected, the baseline underestimates affected users as it does not exploit the knowledge of the targeted removal process. More interestingly, when compared to Fig. 5, we found that the curve corresponding to the estimated affected users by the baseline is almost identical to the coverage curve obtained under a random intervention (i.e., the degree of an injected node being statistically the same as a randomly chosen node in the original network without injected nodes). This suggests again that the baseline works only if the intervention is purely randomized and easily fails when this assumption does not hold.

In summary, our causal inference framework gives a significant improvement upon the structural fidelity of inferred latent networks as a result of properly exploiting the causal influence of targeted interventions in both synthetic and realistic settings. This study recognizes and emphasizes the importance and benefits of a combined learning of network generative process (i.e., network model) and the underlying process that leads to partial network observability (i.e., intervention model).

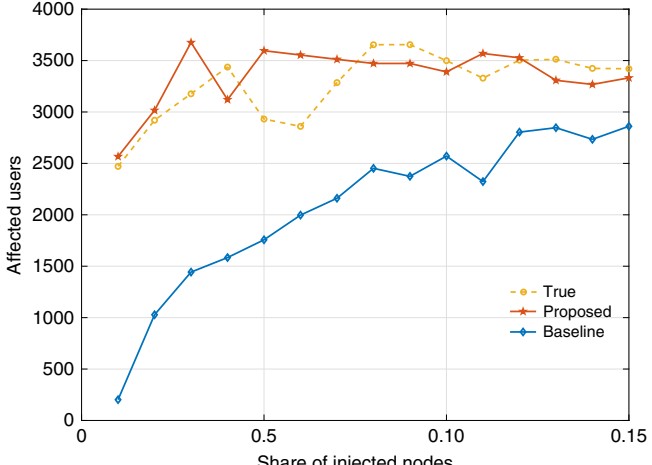

**Fig. 7** Comparison of capability to estimate affected users ($\alpha = 1$)

## Discussion

In sharp contrast to prior work based on random removal assumption, we proposed a causal inference framework that considers the statistical behavior of intervention and its causal influence on reshaping the underlying distribution of the latent structure through a sequence of dynamic attack strategies. While its application to vast domains needs to be further investigated, we demonstrated its effectiveness in three case studies concerning three different sets of real complex networks in

social-, genomic-, and neuro-science. Moving beyond these examples, we believe our framework can help us explore a wide spectrum of real complex networks problems. For instance, instead of assuming the intervention model a priori, we can alternatively consider a range of intervention policies and postulated network scales (e.g., network size) to infer the missing network under different hypotheses. These hypothesized networks can potentially guide us to rediscover the latent structure in a variety of networked systems that are subject to variations and limited observabilities.

On a different direction, as we observed in our experiments, many real networks are following a structural organization principle (e.g., small world property) that is robust against random removals. In biological systems, the prevalence of such structural robustness against random removal might reflect some degree of evolutionary wisdom since it offers protection against internal or external random perturbations and mutations. However, under the exact same principle, adversarial entities like viruses and cancer cells develop their counter-strategies to cancel out these structural advantages to maximize their own survival benefits. This notion also applies to social, computer, and traffic networks where infrequent yet highly connected hubs actually dominate the normal operation of entire systems. They are consequently much more easily targeted and, once sabotaged, give rise to greater social, political, and economic expenses. Real threats and interventions are therefore rarely randomized and an inference framework that admits widely ranged structural interventions is a must. From these perspectives, we would like to see the application of our proposed framework beyond the presented examples and extend its reach to a much broader class of topics.

## Methods

**Taming the interdependency between network inference and adversarial interventions**. To make our discussion concrete without the loss of generality, we consider $\mathcal{G}$ to be universal model that induces a linking probability measure $\mathcal{P}$ where $p_{i,j} \in \mathcal{P}$ quantifies the probability of a link between an arbitrary pair of nodes $i$ and $j$ in the network. In fact, most of state-of-the-art network models including stochastic block model[53], Kronecker graphs[54], and MFNG all fit into this construction. One issue arises under this assumption. Given the partially observed network $G_t$ with its node arbitrarily indexed, we do not know the mapping between a node index $i$ to its associated linking probability measure $i' \in \mathcal{P}$ as in the original network $G_0$. Consequently, we also need to infer such correspondence. Define $\psi$: $V \to N$ to be the mapping between the node index $i$ to its associated linking probability measure index $i' = \psi(i)$. We then rewrite Eq. (1) as follows:

$$\arg\max_{\mathcal{G}, M_t, \psi, \pi} P(G_t, M_t, \psi, \pi | \mathcal{G}, \mathcal{A}) \quad (3)$$

Clearly, Eq. (3) implies an interdependent dilemma: Inferring the missing part of the network requires full knowledge of the underlying model of the original network. Identifying the underlying model calls for full knowledge of the original network. In other words, the optimal solution to Eq. (3) requires the maximization over the generative model $\mathcal{G}$ and the missing information $\{M_t, \psi, \pi\}$ at the same time.

To decouple them, a straightforward approach is to first consider a maximum likelihood estimator (MLE) for the underlying model $\mathcal{G}$ by marginalizing over the missing information $\{M_t, \psi, \pi\}$.

$$\mathcal{G}^* = \arg\max_{\mathcal{G}} P(G_t | \mathcal{G}, \mathcal{A}) = \arg\max_{\mathcal{G}} \int\int\int P(G_t, M_t, \psi, \pi) | \mathcal{G}, \mathcal{A}) dM_t d\psi d\pi \quad (4)$$

where the likelihood $P(G_t, M_t, \psi, \pi | \mathcal{G}, \mathcal{A})$ can be calculated as follows:

$$P(G_t, M_t, \psi, \pi | \mathcal{G}, \mathcal{A}) = (\Pi_{(i,j) \in E_0} p_{\psi(i), \psi(j)} \Pi_{(i',j') \notin E_0} (1 - p_{\psi(i'), \psi(j')})) * \gamma \Pi_{s=0}^{t-1} A_\alpha(d(\pi^{-1}(s)), s) \quad (5)$$

where $\pi^{-1}(s) = \Delta v(s)$ represents the node removed at time $s \in [0, t-1]$ and $d(\pi^{-1}(s))$ denotes its degree. The first two terms represent how likely the network structure is entailed by the underlying model. The third term encodes how much the inferred sequence of missing substructures can be explained by the statistical behavior of the attacker. The discount factor $\gamma$ reflects the disagreement between the attacker's structure preference of its target and what network model suggests. If the intervention is hub-prioritized whereas the underlying network model discourages highly connected nodes, the discount factor is consequently large to emphasize the influence of the intervention. Otherwise, a small discount factor is selected. In a special case when the intervention is purely randomized (i.e., $\alpha = 0$),

the discount factor $\gamma$ is 0. Only in this case the formulation in Eq. (3) can be reduced to the well-researched network completion problem.

Solving the model identification problem in Eq. (4) simultaneously leads us to the solution of the inference problem. However, the marginalization over the latent variable $M_t$, $\psi$, and $\pi$ is computationally intractable. To approach this problem, we replace the marginalization process by constructing a series of maximization steps over the incomplete likelihood function $P(G_t, M_t, \psi, \pi | \mathcal{G}, \mathcal{A})$ conditioned on the propagated belief about the model parameters $\mathcal{G}$.

Formally at $i$th step by taking the log-likelihood,

$$Q(\mathcal{G}|\mathcal{G}^{(i)}) = \int \log[P(G_t, M_t, \psi, \pi | \mathcal{G}, \mathcal{A})] P(\psi, M_t, \pi | \mathcal{G}^{(i)}, G_t) dM_t d\psi d\pi \quad (6)$$

$Q(\mathcal{G}|\mathcal{G}^{(t)})$ constructs an incomplete maximum likelihood function in terms of observable part of the network. It averages out the contribution of the missing information $\{M_t, \psi, \pi\}$ by using the incomplete MLE for $\mathcal{G}$ at previous step to infer a current guess on $\{M_t, \psi, \pi\}$. Complex models for high dimensional data lead to intractable integrals as the ones in Eq. (6). To overcome this drawback, we approximate the integral conditioned on the current guess on the generative measure $\mathcal{G}$ via a Monte-Carlo sampling procedure,

$$Q(\mathcal{G}|\mathcal{G}^{(i)}) = \lim_{K \to \infty} \frac{1}{K} \sum_i^K \log[P(G_t, M_t^{(i)}, \psi^{(i)}, \pi^{(i)} | \mathcal{G}, \mathcal{A})] \quad (7)$$

where the samples are drawn from $P(\psi, M_t, \pi | \mathcal{G}^{(i)}, G_t)$. Update the estimator of $\mathcal{G}^{(i)}$ by maximizing $Q(\mathcal{G}|\mathcal{G}^{(t)})$,

$$\mathcal{G}^{(i+1)} = \arg\max_{\mathcal{G}^*} Q(\mathcal{G}|\mathcal{G}^{(i)}) \quad (8)$$

Under regularity conditions[55,56] and given a suitable starting value $\mathcal{G}^{(0)}$, the resulting sequence $\mathcal{G}$ will converge to a local maximizer of the likelihood function by alternating the above procedure until the difference $P(G_t|\mathcal{G}^{i+1}, \mathcal{A}) - P(G_t|\mathcal{G}^i, \mathcal{A})$ changes by an arbitrarily small amount. It should be noted that the above procedure not only constructs a MLE for the underlying model $\mathcal{G}$ but also simultaneously returns the most probable guess on $M_t$, $\psi$ and $\pi$ in a maximum likelihood sense.

The finite sum approximation of the expectation depends on being able to draw samples from the joint distribution $P(G_t, M_t, \psi, \pi | \mathcal{G}, \mathcal{A})$. Instead of using uniform sampling that generates unimportant samples in an unprincipled fashion, we need to confine the samples to be drawn from the region where the integrand of Eq. (6) is large. Moreover, the computational intractability of sampling the posterior joint distribution also originates from the factorial dependence of the sample space on the size of the original network and the missing network. This factorial dependence comes from the requirement to infer the time-stamp mapping $\pi$ and the linking probability measure mapping $\psi$ for each node in the missing network $M_t$. Consider a temporally ordered sequence of subgraph $Z_t = \{z_0, z_1, ..., z_{t-1}\}$ that corresponds to trajectory of the subgraph removed at each step of the intervention up to time $t$. Inferring the optimal $\pi$ and $\psi$ for each node implies that when maximizing Eq. (5) the following relation holds:

$$\forall j \in V(M_t), \exists z_i \in Z_t, \pi(j) = V(z_i) \vee \forall i, j \in V(M_t), \pi(i) = \pi(j) \Leftrightarrow i = j \quad (9)$$

$$\forall j \in V(M_t), \exists j' \in \mathcal{P}, \psi(j) = j' \quad (10)$$

Note that $V(G_{k+1}) = V(G_k)/V(z_k)$ and $E(G_k) = E(G_{k-1}) \backslash \{e_{i, V(z_{k-1})} | \forall i \in V(G_{k-1})\}$. Equation (9) implies that the size of the sample space is given by the number of all possible permutations of the time stamps $|M_t|!$, hence the need for factorially many samples for the finite sum approximation (7) to be valid. One key observation here is that $Z_t$ is also a sufficient statistic for the incomplete likelihood function $Q$ in terms of $\{M_t, \pi\}$. In other words, we can eliminate the need to infer $\pi$ and $\psi$ separately by introducing the following mapping $\psi' : Z \to \mathcal{P}$ that satisfies,

$$\psi'(\pi(i)) = \psi(i) \quad (11)$$

Note that $M_t = \cup_{z_i \in Z_t} \{z_i\}$ thus the log-likelihood function in Eq. (6) can be reduced as,

$$\begin{aligned} Q(\mathcal{G}|\mathcal{G}^{(i)}) &= \int \log[P(G_t, M_t, \psi, \pi | \mathcal{G}, \mathcal{A})] P(\psi, M_t, \pi | \mathcal{G}^{(i)}, G_t) dM_t d\psi d\pi \\ &= \int \log[P(G_t, Z_t, \psi' | \mathcal{G}, \mathcal{A})] P(\psi', Z_t | \mathcal{G}^{(i)}, G_t) dZ_t d\psi \\ &= \lim_{K \to \infty} \frac{1}{K} \sum_i^K \log[P(G_t, Z_t^{(i)}, \psi'^{(i)} | \mathcal{G}, \mathcal{A})] \end{aligned} \quad (12)$$

The transformation in Eq. (12) suggests that we just need to infer the transition path $Z_t$ and the linking measure assignment $\psi'(V(z_k))$ (instead of $\psi$) as in MFNG for each subgraph $z_k \in Z_t$. Alternatively stated, the nodes in $M_t$ are anonymized and their mapping to $Z_t$ is not important given the knowledge of $\psi'$. To efficiently estimate the joint distribution $P(\psi', Z_t | \mathcal{G}^{(j)}, G_t, \mathcal{A})$, we choose to construct a Monte Carlo Markov Chain (MCMC) that alternates sampling from $P(Z_t | \psi'^{(\tau-1)}, \mathcal{G}^{(j)}, G_t, \mathcal{A})$ and $P(\psi' | Z_t^{(\tau)}, \mathcal{G}^{(j)}, G_t, \mathcal{A})$. However, MCMC offers only a sketch of sampling schedule guaranteeing that the drawn samples follow asymptotically $P(\psi', Z_t | \mathcal{G}^{(j)}, G_t, \mathcal{A})$. The overall complexity of this schedule still depends on how efficiently the samples can be taken from the individual conditional distributions. For sampling the permutations of $P(\psi' | Z_t^{(\tau)}, \mathcal{G}^{(j)}, G_t, \mathcal{A})$,

many existing strategies are applicable based on, to name a few, the construction of a MCMC[54] or a simulated-annealing type swapping method[26]. We focus on the design of sampling for $P(Z_t|\psi'^{(\tau-1)}, \mathcal{G}^{(j)}, G_t, \mathcal{A})$ as there exists a nicely recursive optimal substructure that is very similar to the most probable sequence problem in Markov decision process and hidden Markov model (HMM). We take advantage of this recursive structure and draw samples from $P(G_{0:t-1}|\psi'^{(\tau-1)}, \mathcal{G}^{(j)}, G_t, \mathcal{A})$ efficiently via a combination of rejection sampling and Metropolis sampling.

**Decouple the sampling of joint distribution** $P(\psi', Z_t|\mathcal{G}^{(j)}, G_t)$. Formally, by proper choice of an acceptance criteria $A(s^*, s)$ and a proposal transition distribution $q(s^*|s)$ to satisfy the detailed balance condition,

$$p(s)p(s^*|s) = p(s^*)p(s|s^*) \quad (13)$$

where $p(s^*|s) = A(s^*, s)q(s^*|s)$. It follows that the Markov chain $\{s^{(i)}\}$ defined by $q(s^*|s)$ has a stationary distribution of $p(s)$. By restricting the proposal transition only from $s = \{s_{/k}, s_k\}$ to $s^* = \{s_{\backslash k}, s_k^*\}$ for $\forall k$ with following acceptance probability:

$$A(s^*, s) = \min\left(1, \frac{p(s^*)q(s|s^*)}{p(s)q(s^*|s)}\right) \quad (14)$$

where $s_{\backslash k}$ denotes all but the $k$th component. The joint distribution $p(s)$, as the stationary distribution of this constructed Markov chain, can then be sampled by cycling through separate sampling procedures from the $k$th conditional distribution $p(s_k|s_{\backslash k})$ for all $k$'s. This special case of MCMC sampling provides us an efficient way to decouple the sampling of $Z_t$ and $\psi'$. The algorithmic details are stated as follows.

- Denote $B$ as the number of Burn-in samples, $K$ as the total number of samples, and $S = \{Z_t^{(i)}, \psi'^{(i)}\}$ to be the set of samples drawn from $P(\psi', Z_t|\mathcal{G}^{(j)}, G_t, \mathcal{A})$
- Initialize $\{Z_t^{(0)}, \psi'^{(0)}\}$ and set $S = \varnothing$.
- Repeat the following Steps (4)–(6) for all $\tau < K + B$.
- Sample $Z_t^{(\tau)} \sim P(Z_t|\psi'^{(\tau-1)}, \mathcal{G}^{(j)}, G_t, \mathcal{A})$;
- Sample $\psi'^{(\tau)} \sim P(\psi'|Z_t^{(\tau)}, \mathcal{G}^{(j)}, G_t, \mathcal{A})$;
- Add $\{Z_t^{(\tau)}, \psi'^{(\tau)}\}$ to $S$ if $\tau < B$.

**Optimal recursive structure in** $P(Z_t|\psi'^{(\tau-1)}, \mathcal{G}^{(j)}, G_t, \mathcal{A})$. To sample $P(Z_t|\psi'^{(\tau-1)}, \mathcal{G}^{(j)}, G_t, \mathcal{A})$, we notice the transition equation $G_{k+1} = G_k/z_k$ holds for $\forall z_k \in Z_t$. Denote $G_{0:t-1} = \{G_{t-1}, G_{t-2}, ..., G_0\}$ as an ordered sequence of residual graph after each intervention up to time $t - 1$ such that,

$$G_{0:t-1}\backslash G_t = \{\cup_{k=1}^{i} z_{t-k}\}_{i=1,2,...,t} \quad (15)$$

Given $G_t$, this relation suggests the knowledge of $Z_t$ and $G_t$ is interchangeable and the following probability are identical under the transformation given in Eq. (15)

$$P(Z_t|\psi'^{(\tau-1)}, \mathcal{G}^{(j)}, G_t, \mathcal{A}) = P(\cup_{k=0}^{t-1}\{G_k + 1\backslash G_k\}|\psi'^{(\tau-1)}, \mathcal{G}^{(j)}, G_t, \mathcal{A})$$
$$= P(G_{0:t-1}|\psi'^{(\tau-1)}, \mathcal{G}^{(j)}, G_t, \mathcal{A}) \quad (16)$$

by Bayesian rule,

$$P(G_{0:t-1}|\psi'^{(\tau-1)}, \mathcal{G}^{(j)}, G_t, \mathcal{A}) = \beta P(G_t|G_{0:t-1}, \psi'^{(\tau-1)}, \mathcal{G}^{(j)}, \mathcal{A})P(G_{0:t-1}, \psi'^{(\tau-1)}, \mathcal{G}^{(j)}, \mathcal{A}) \quad (17)$$

notice the transition of $G_k$ is driven by the attacker that depends only on the network configuration presented to it at the time of the intervention. In other words, the transition is Markovian and conditionally independent of the network model, hence we have,

$$P(G_{0:t-1}|\psi'^{(\tau-1)}, \mathcal{G}^{(j)}, G_t, \mathcal{A})$$
$$= \beta P(G_t|G_{t-1}, \mathcal{A})P(G_{t-1}|\psi'^{(\tau-1)}, \mathcal{G}^{(j)}, \mathcal{A})\{P(G_{0:t-2}|\psi'^{(\tau-1)}, \mathcal{G}^{(j)}, G_{t-1}, \mathcal{A})\} \quad (18)$$

where $\beta$ is the appropriate normalization factor. $P(G_{0:t-1}|\psi'^{(t-1)}, \mathcal{G}^{(j)}, G_t, \mathcal{A})$ quantifies the probability of a sequence of interventions up to time $t$ given the underlying network and adversarial attack models. $P(G_t|G_{t-1}, \mathcal{A})$ represents the transition model determined by the adversarial intervention (as it is the only driver of the transition). $P(G_{t-1}|\psi'^{(\tau-1)}, \mathcal{G}^{(j)}, \mathcal{A})$ considers how likely $G_{t-1}$ can be explained by the underlying network model. Given $G_{t-1}$, the product of first two quantifies how likely $G_{t-1}$ explains the transition of a network (i.e., $G_{t-1} \to G_t$) described by the model $\mathcal{G}$ under the adversarial intervention $\mathcal{A}$. It hints on that the guess we take on $G_{t-1}$ from our observation on $G_t$ should be supported by both the adversarial intervention model and the network model, which emphasizes again the necessity of a combined knowledge of network and adversarial intervention models. As a result, the prior methods that consider only the network models cannot be applied here.

More importantly, we note that the third term $P(G_{0:t-2}|\psi'^{(\tau-1)}, \mathcal{G}^{(j)}, G_{t-1}, \mathcal{A})$ is exactly a sub-problem of original one, hence suggesting a nice recursive structure of the inference problem, which resembles the most likely sequence problem in HMM. In principle, such recursive optimal problem structure immediately implies a dynamic programming (e.g., Viterbi algorithm) that solves the problem optimally given the initial distribution on $G_0$ if $\psi'^*$ and $\mathcal{G}^*$ are known. If not, we instead take advantage of this recursive structure and draw samples from

$P(G_{0:t-1}|\psi'^{(\tau-1)}, \mathcal{G}^{(j)}, G_t, \mathcal{A})$. More precisely, for each subgraph $G_s$ in time, we recursively sample $G_s^{(\tau)}$ from $P(G_s|\psi'^{(\tau-1)}, \mathcal{G}^{(j)}, \mathcal{A})$ and accept it with a probability $A(G_s)$ conditioned on the previously drawn sample $G_{s+1}$,

$$A(G_s^{(\tau)}) = \frac{f(G_s^{(\tau)}; G_{s+1}^{(\tau)})}{P(G_s^{(\tau)}|\psi'^{(\tau-1)}, \mathcal{G}^{(j)}, \mathcal{A})} \quad (19)$$

Therefore, the probability to accept $G_s^{(\tau)}$ is $f(G_s^{(\tau)}; G_{s+1}^{(\tau)})$ and the probability $A(G_{0:t-1}^{(\tau)})$ to accept the entire path $G_{0:t-1}$ is given by,

$$A(G_{0:t-1}^{(\tau)}) = \Pi_{s=0}^{t-1} f(G_s^{(\tau)}; G_{s+1}^{(\tau)})$$
$$= P(G_{0:t-1}^{(\tau)}|\psi'^{(\tau-1)}, \mathcal{G}^{(j)}, G_t, \mathcal{A}) \quad (20)$$

The second equality holds due to the recursive structure in Eq. (18). One straightforward sampling method is rejection sampling that takes samples exactly from the target distribution given a proper proposal distribution. Fortunately, such a proposal distribution can be naturally constructed by $P(G_s|\psi', \mathcal{G}_k^{(j)}, \mathcal{A})$ in the recursive structure of our problem and it is always locally lower bounded by $f(G_s; G_{s+1})$ (hence being overall lower bounded by $P(G_{0:t-1}|\psi'^{(\tau-1)}, \mathcal{G}^{(j)}, G_t, \mathcal{A})$). Note that a strict ordering holds for $G_{0:t-1}$ such that $G_i \subset G_j$ for $\forall i > j$. Therefore, sampling $P(G_s|\psi', \mathcal{G}_k^{(j)}, \mathcal{A})$ requires only the sample on $z_s = G_s/G_{s+1}$. Algorithmically, the following procedure states the sampling process:

- For $s = t - 1$ to 0, repeat the following Steps (2) and (3) until $a_s < = f(G_s^{(\tau)}; G_{s+1}^{(\tau)})$
- Draw a sample $G_s^{(\tau)}$ from $P(G_s|\psi'^{(\tau-1)}, \mathcal{G}^{(j)}, \mathcal{A})$.
- Draw a sample $a_s$ from $U(0, P(G_s^{(\tau)}|\psi'^{(\tau-1)}, \mathcal{G}^{(j)}, \mathcal{A}))$.

The above procedure provably produces samples from $P(Z_t|\psi'^{(\tau-1)}, \mathcal{G}^{(j)}, G_t)$ whereas the acceptance rate can be practically low during the experiment as a result of unprincipled sampling from unimportant regions (low probability) of $P(G_s|\psi'^{(\tau-1)}, \mathcal{G}^{(j)}, \mathcal{A})$. To conquer this, we supplement it with the construction of a Markov chain such that we can efficiently draw samples from $P(Z_t|\psi'^{(\tau-1)}, \mathcal{G}^{(j)}, G_t, \mathcal{A})$ once a sample $Z^{(\tau)}$ is obtained by the above procedure. Specifically, given $Z^{(\tau)} = \{z_{t-1}^{(\tau)}, z_{t-2}^{(\tau)}, ..., z_0^{(\tau)}\}$, we define the transition probability for each $z_k^{(\tau)}$ by,

$$P_{z_k^{(\tau)}|z_k^{(*)}} = \frac{1}{d(i)} \frac{p_{i,x}}{\sum_y p_{i,y}} \quad (21)$$

where $i \in V(z_k^{(\tau)})$, $x \in V(z_k^{(*)})$ and $y \in V\left(G_t \cup \{z_{t-1}^{(\tau)}\}, z_{t-2}^{(\tau)}), ..., z_{k+1}^{(\tau)}\right)$. For $\forall k < t$, the following procedure induces a Markov chain with respect to $z_k$ with its stationary distribution being $f(G_k; G_{k+1})$:

- Randomly sample an edge $(i, j)$ where $i \in V(z_k^{(\tau)})$ and $j \in V\left(G_t\{z_{t-1}^{(\tau)}\}, z_{t-2}^{(\tau)}), ..., z_{k+1}^{(\tau)}\}\right)$ with a probability $P\{(i, j)\} = 1/d(i)$.
- Rewire $(i, j)$ to $(i, j')$ to produce $z_k^{(*)}$ with probability, $p_{i,j'} / \sum_y p_{i,y}$ where $y \in V\left(G_t\{z_{t-1}^{(\tau)}\}, z_{t-2}^{(\tau)}), ..., z_{k+1}^{(\tau)}\}\right)$.
- Accept $z_k^{(*)}$ with probability $A(z_k^{(*)}, z_k^{(\tau)}) = min\left(1, \tilde{p}(z_k^{(*)})P_{z_k^{(*)}|z_k^{(\tau)}}/\tilde{p}(z_k^{(\tau)})P_{z_k^{(\tau)}|z_k^{(*)}}\right)$

where $\tilde{p}(z_k) = f(G_k; G_{k+1})$. Define $\tilde{P}_{z_k^{(*)}|z_k^{(\tau)}} = P_{z_k^{(*)}|z_k^{(\tau)}}A(z_k^{(*)}, z_k^{(\tau)})$., it can be easily shown that the constructed Markov chain satisfies the following detailed balance condition,

$$f(G_k^{(\tau)}; G_{k+1}^{(\tau)})\tilde{P}_{z_k^{(\tau)}|z_k^{(*)}} = f(G_k^{(*)}; G_{k+1}^{(*)})\tilde{P}_{z_k^{(*)}|z_k^{(\tau)}} \quad (22)$$

Constructing the above Markov chain for each component in $Z^{(\tau)}$, it follows Eq. (20) that samples are drawn from $P(Z_t|\psi'^{(\tau-1)}, \mathcal{G}^{(j)}, G_t, \mathcal{A})$.

**A MCMC approach to sample** $P(\psi'|Z_t^{(\tau)}, \mathcal{G}^{(j)}, G_t, \mathcal{A})$. We construct a Markov chain for the sampling of mapping $\psi'$ by repeating the following procedure:

- Randomly sample two indexes $i$ and $j$ in $\psi'^{(\tau)}$ and swap them to obtain $\psi'^{(*)}$.
- Accept $\psi'^{(*)}$ with probability $A(\psi'^{(*)}, \psi'^{(\tau)})$.

where $A(\psi'^{(*)}, \psi'^{(\tau)})$ is defined by,

$$A(\psi'^{(*)}, \psi'^{(\tau)}) = \min\left(1, \frac{P(\psi'|Z_t^{(*)}, \mathcal{G}^{(j)}, G_t)}{P(\psi'|Z_t^{(\tau)}, \mathcal{G}^{(j)}, G_t)}\right) \quad (23)$$

**Optimization strategy to maximize** $Q(\mathcal{G}|\mathcal{G}^{(i)})$ **under MFNG model** $\mathcal{G}_k$. We adopt a batch gradient descent approach to optimize the incomplete log-likelihood function $Q(\mathcal{G}_k^{(j+1)}|\mathcal{G}_k^{(j)})$ at $j$th iteration with the following procedure:

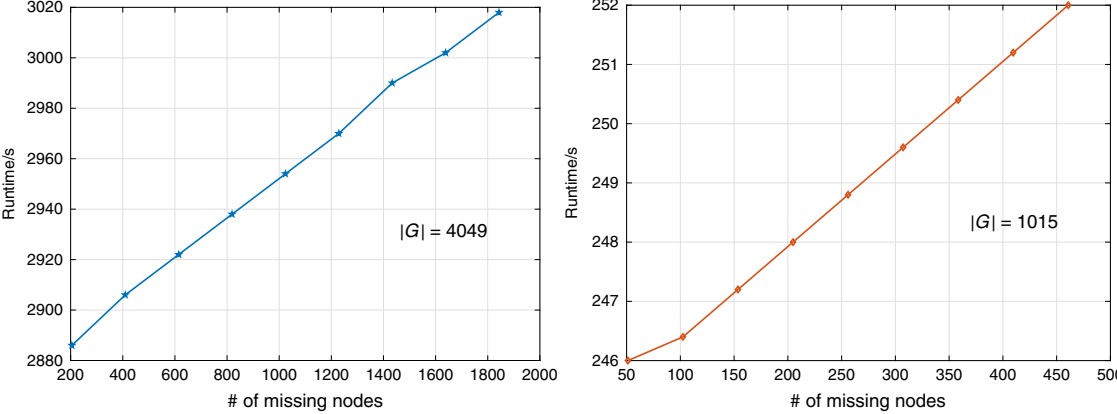

**Fig. 8** Inference runtime as a function of missing nodes with a network of size 4049 and 1015

- $\{Z_t^{(\tau)}, \psi^{(\tau)}\} \leftarrow P(\psi', Z_t | \mathcal{G}_k^{(j)}, G_t, \mathcal{A})$
- $l^{(\tau)} = \log[P(G_t, Z_t^{(\tau)}, \psi'^{(\tau)} | \mathcal{G}^{(j)_k}, \mathcal{A})]$
- $g^{(\tau)} = \partial l^{(\tau)} / \partial \mathcal{P}_0$
- Repeat Steps (1)–(3) for $\tau = B + 1$ to $K + B$
- $\mathcal{P}_0^{(j+1)} = \mathcal{P}_0^{(j)} + \sigma/K * \sum_\tau g^{(\tau)}$
- $\mathcal{G}_k^{(j+1)} = (m, k, \mathcal{P}_0^{(j+1)}, \mathcal{L})$

The derivation of the gradient of the log-likelihood function can be found in Supplementary Note 5.

**Computational complexity**. Overall, E step involves taking samples from distribution $P(\psi', Z_t | \mathcal{G}^{(j)}, G_t)$ which can be addressed by the proposed alternated MCMC sampling processes for both $P(Z_t | \psi'^{(\tau-1)}, \mathcal{G}^{(j)}, G_t, \mathcal{A})$ and $P(\psi' | Z_t^{(\tau)}, \mathcal{G}^{(j)}, G_t, \mathcal{A})$. Since MCMC surely produces a sample after each iteration in $O(1)$ time, the amortized sampling cost is thus $O(|Z_t|)$ where $|Z_t|$ being the size of latent network. Consider $K + B$ samples in total, each iteration of E step takes $O((K + B)|Z_t|)$. M step involves the optimization of the $Q$ function by gradient descent. The amortized cost of gradient calculation is given by $O(|E_0|)$ per sample (see S5). Therefore, the worst-case computational complexity of one iteration of EM is $O(KS|E_0| + (K + B)|Z_t|)$ where $S$ is the number of optimization steps, $K$ is the number of samples and $B$ is the number of burn-in samples. Note that $|E_0|$ is a quadratic function of network size in the worst case. Thus, $O(KS|E_0| + (K + B)|Z_t|) = O(KS|E_0|)$ and the computational complexity is mainly decided by the M-step.

**Method constraints**. There are several key aspects that could be improved by our future work. While the assumptions made in the attack model seem plausible, the real attacks may not follow a consistent statistical pattern as the one described in Eq. (2). For instance, the causal structure of the attacking sequence considered in our framework can be more sophisticated by the coordination/interaction (sharing of information) among multiple attackers co-existing in the network. Attackers may not necessarily operate under the same strategy, which makes it challenging to construct consistent and accurate models to characterize their behavior. Consequently, it is important to incorporate attack strategies as part of the network inference framework (e.g., either estimating the unknown parameters of the attack model together with the network model in an EM approach or estimate them separately based on additional information when available). Since real networks can change their growth rules and possibly their self-similar structure over time (e.g., co-existence or emergent transition of small-world and multi-fractal scaling observed in complex networks[57,58]), a generative model that captures all the structural features of interested networks can be difficult to build. As a result, applying the proposed method to retrieve the latent subnetwork resulted from attacks on real-world networked systems (e.g., social network manipulation and intervention) requires time-labeled data collection process. This data collection should enable reliable identification of and estimation on the statistical behavior of attackers and its variations over time (e.g., through multiple piece-wise temporal windows that correspond to different statistical modes/patterns of the attacker). Towards this end, an integration of continuous anomaly detection and data monitoring system is a must to interface with the proposed framework and other analytical tools (e.g., opinion diffusion dynamics) for identification, influence assessment, and source tracking of the adversarial interventions on real-world networks.

Finally, applying the inference framework to large scale networks would require more efficient computational techniques. As detailed in the "Methods" section, the overall computation complexity of one EM iteration is $O(KS|E_0|)$ where $|E_0|$ is the number of links in the original network, $K$ is the number of samples and $S$ denotes optimization steps. In the worst case, $|E_0|$ is a quadratic function of network size

and the number of samples required to identify the network model also grows exponentially. This can be shown in Fig. 8 that runtime is dominated by the network size and slowly increases as the $|Z_t|$ grows where $|Z_t|$ is the number of latent nodes. The algorithm is written in Matlab and runs on i7-4790K with 32 GB memory where $K = 40,000$, $B = 10,000$ and $S = 10$.

While extending the inference framework to larger scales requires further work, we also need to be very cautious about the interpretation of the worst-case computational complexity. Firstly, many real networks are sparse, which makes the runtime of proposed algorithm run much faster than the worse-case computational complexity implies. Secondly, the size of many biological networks varies from a few hundreds to a few thousands of nodes, which makes the proposed framework suitable for use and further extension to specific biological investigations. Thirdly, social networks are known to possess small world and scale-free properties, as well as rich in the degree of locality (related to occupation, age, or geographic proximity). Also, attackers can hardly grasp the global information about the networks. This means that a targeted attack usually happens to a localized subnetwork (observable part of the network for the attacker) rather than the entire network. Combining these important aspects with more realistic attack strategies and opinion/information diffusion models opens up a rich yet challenging class of network reconstruction and inference problems for the network science research community.

## Data availability

The Facebook social network can be accessed from Stanford Network Analysis Project (SNAP) [http://snap.stanford.edu]. The hu.Map network can be accessed from Biological General Repository for Interaction Datasets (BioGRID 3.5) [https://downloads.thebiogrid.org/BioGRID/Published-Datasets/Marcotte2017]. The human brain consensus connectome can be found in Budapest Reference Connectome 3.0 [https://doi.org/10.1016/j.neulet.2015.03.071]. The source codes for generating the results can be found at Github [https://github.com/urashima9616/NetworkReconstruction]. Supplementary Note 6 gives a detailed instruction to run the code.

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

## Acknowledgements
The authors gratefully acknowledge the support by the U.S. Army Defense Advanced Research Projects Agency (DARPA) under Grant no. W911NF-17-1-0076, DARPA Young Faculty Award under Grant no. N66001-17-1-4044, and the National Science Foundation under CAREER Award CPS/CNS-1453860 support. The views, opinions, and/or findings contained in this article are those of the authors and should not be interpreted as representing the official views or policies, either expressed or implied, of the Defense Advanced Research Projects Agency, the Department of Defense or the National Science Foundation.

## Author contributions
Y.X. and P.B. conceived the research and designed the analysis. Y.X. designed the algorithm and developed the computational tools. Y.X. and P.B. conducted the analysis and wrote the manuscript.

## Additional information

**Competing interests:** The authors declare no competing interests.

