## [peer review file · Nature Communications]

Reviewer #1 (Remarks to the Author):

Dear authors,

First of all, thank you for the detailed and matching response to my comments. I very much enjoyed the direct and bona fide debate. I respond further only to the issues which I believe to be outstanding.

C-1. I agree that whilst real systems are dirty and may not fit a certain stable probabilistic structure, it doesn't mean there cannot be a useful abstraction. Certainly some of the references given in the response seems to indicate that there can be a degree of consistency in attack behaviour. On a more philosophical point, one may wish to see if there is a need to distinguish network formation laws (e.g. a decentralised complexity) with attacks (which maybe a form of centralised complexity). What would be useful is to perhaps set out 1-2 mechanics for real-world attacks that obey a certain pattern in the SI. This would help to motivate readers and also expand on the impact of the paper.

F. I would suggest also pointing out where this approach might not work in the paper (which also relates to some C-1 points). Often knowing its limitations is very helpful for end users and also enables others to follow up on further research. It maybe the case that limitations are better demonstrated with simpler examples (which chimes with another reviewer's comments I believe).

H. I am very happy that the paper is better worded and when adding in my recommendation for setting clear limits for the graph and attack structure boundaries (see C-1 and F), I think it makes a solid and impactful contribution to this body of important literature.

In summary, I recommend an accept (subject to minor changes recommended above).

Best

Reviewer #2 (Remarks to the Author):

This is a follow up on my first review (I was reviewer 2). I read the reply to my comments carefully. If I am trying to be witty, I would say that the problem with "Catching the invisible" is that the authors want to "have their cake and eat it too".

The authors argue in their reply that their manuscript is providing a first step on a new approach to uncover the presence and impact of hidden, antagonistic networks of agents. This is a good and important goal. However, then authors then write their manuscript as if they have made a much greater advance than they actually achieved. This is my problem!

The paper is written as if their approach is going to make visible what happened in the 2016 US Presidential Elections and the 2016 Brexit Referendum. This is extremely misleading and something that I hope Nature Communications will not condone.

If the authors want to write an serious paper, then they should consider several network models of increasing complexity and several attacking strategies of increasing complexity, and several attempts at guessing the parameters of the attacks. And then, they should write about the conditions under which one can or cannot "catch the invisible".

As is, the manuscript is a misleading exercise whose aim appears to be getting press attention.

Reviewer #3 (Remarks to the Author):

The authors have tried to address all my issues and to improve the presentation of their work.

In particular, adding Fig. 1 in the main text with an introductory example made the article clearer.

I also appreciate the addition of the part about the algorithmic complexity in the Supplementary Material.

I think that the novel method proposed in this work represents an important advance in the field of network reconstruction and could also be of interest for a broader audience dealing with complex network and may be suited for publication in Nature Communications.

However, I think the current version of the manuscript should still be improved.

In particular, the authors could do a better job of taking into account the issues raised by the referees and integrating their answers in the manuscript.

Firstly, it is difficult to understand what changes were done to the new version of the manuscript since they do not indicate it clearly. They could have provided a version of the article with the

changes highlighted and reported all the changes and their location in the answer to the referees letter (they did it for some changes but not consistently).

Even if this is, in a sense, a new submission (since it was transferred from Nature), this would have made their work much clearer.

Concerning their answers to my questions:

- About the example of the election manipulation:

I appreciate the clearer introduction of this example on page 6. However I still think that this example is misleading.

I agree that, in a very idealized model of election manipulation, their method may be used to "discover the hidden subnetwork of attackers who try to manipulate people's opinion with a social network", but I don't see how it could be used to measure the real influence of the attackers on the vote in a real world scenario. While they addressed some of my remarks concerning this point in the answer to the referees letter, they did not include the remarks about the limitations of their method in the article.

If they persist in wanting to use this case as an example of application of their method, they need to clearly explain which questions it can answer and which it cannot. They need to stipulate that their simulation is highly idealized and clarify the limitations of this example in order to not mislead readers into thinking that their approach solves the problem of opinion manipulation as this is a much more complex problem than what they simulate.

They use the case of Cambridge Analytica to justify their example, saying "We simulate a similar opinion manipulation

attack as the reported case of Cambridge Analytica on Facebook users with the difference being that we assume there exists a

multitude of running agents that try to influence voting decisions."

However, the case of Cambridge Analytica is not very similar to their scenario. The app used by Cambridge Analytica was only used to gather information on people using it (and on their friends) in order to build psychological profiles.

This was then used to target users based on their personality traits with digital ads and fundraising appeals (see <https://www.nytimes.com/2018/03/17/us/politics/cambridge-analytica-trump-campaign.html>).

Comparing their simulation to the case of Cambridge Analytical shows that a realistic simulation should in fact include a targeting that depends on psychological profiles and influence outside of just the neighbors of a users (ads).

I think this is what the author mean in the answer to referee letter, when they say "Another important note here is that there is no influence propagation or opinion/rumor spreading in this type of social network interventions. They can be triggered by the actions performed by the data firms after the attack but they are not participating in it through the "agent apps" by themselves." But this is not precised in the main text.

Their scenario is closer to the case of opinion manipulation using bots which I don't think we have proof that Cambridge Analytical used. If I understand correctly, this is what they mean by automated apps in the answer to referee letter.

They should clarify this point in the manuscript.

For an empirical investigation of opinion manipulation with bots, see for example : Ferrara, Emilio, et al. "The rise of social bots." Communications of the ACM 59.7 (2016): 96-104 and Bessi, Alessandro, and Emilio Ferrara. "Social bots distort the 2016 US Presidential election online discussion." (2016).

Their explanation of the simulation could also be improved. For example, they speak about "the fraction of population being biased" (also in Figure 5) without clearly defining what they consider a biased user (is it the neighbors of the planted influencers?).

What is really unclear for me is how they can measure the "number of biased vote" (Figure 7) and at same time say that "there is no influence propagation or opinion/rumor spreading in this type of social network interventions". If they measure the number of biased vote, they assume that the opinion of the voters was influenced. Otherwise, what they measure may be better describe as the number agent passively collecting data? or the number of user potentially influenced?

But in this case, they need to specify that we have no idea how successful the opinion manipulation is, which means that their result is of little practical use.

This sentence: "Last but not least, we report the estimated votes potentially influenced by the agents who directly interacted with people. This represents a lower bound of biased votes without including the magnifying effect of opinion propagation." is also particularly misleading. "The number of votes potentially influenced" is not equivalent to the number of "biased vote", since we don't know how successful the opinion manipulation is. A potentially influenced vote is not necessarily a biased vote. Not everybody will change their opinion when they are in contact with a opinion manipulator. So the number is in fact an upper bond on the number of biased votes (without taking into account the possibility of opinion propagating farther than to immediate neighbors).

I would appreciate if the authors could clarify this point in the main text.

They also need to discuss in the main text the fact that, as they recognize in their answer to the referees, their model does not take into the possibility that several actors could be competing for influencing the outcome of the elections, which is certainly the case in any real example.

Also in the text we read "With the agents being 1% of the population when $\alpha = 1$, 61.28% of population is already covered", but in Fig. 5 the proportion seems to be rather just below 50% for $\alpha=1$ and a share of spreaders of 0.01. Unless I am misreading the figure.

The value of alpha used in Figs 6 and 7 should be mentioned in the caption (I think it's 1).

- Concerning the issue of algorithmic complexity and running time:

I appreciate the answer of the referee and the supplementary material they added. However, I don't see any reference to this discussion in the main text (apart for a short sentence at the very end of the manuscript).

I am not blaming the authors for not running their algorithms on more powerful machines, but I think they should mention in the main text up to which order magnitude of network size their framework could reasonably be applied. This has direct implication on the type of problems that can be solved with their framework.

- For questions C-6, C-7, C-8 and C-9:

I thank the authors for their answer, but again, a small mention of these points in the main text would improve the manuscript for readers that may have the same questions.

Finally, there still are some inconsistencies in the manuscript.

For example:

p.3 "As a case study, we propose to employ the multi-fractal network generative (MFNG) model (see Supplementary Material S6) as the underlying network model." "S6" Should be S4.

eq. 17 still uses alpha while eq. 18 uses beta.

eq. 6 the differential is still missing.

- Concerning the ability of a researcher to reproduce the work:

The authors used publicly available datasets and provide the source codes for generating the results on github. This is very nice except for the fact that there is very little explanation on how to run the code.

We appreciate again all the comments and suggestions from the anonymous reviewers. We believe all the interactive discussions in the reviewing process greatly help us enhance our manuscript and recognize the key things to improve in both our current work and future research. We would like to present our sincere gratitude to all reviewers for their efforts.

Thank you very much for your consideration,
Yuankun Xue and Paul Bogdan

To avoid the confusion when pointing to the figures in our reproduced changes from the main text, we refer them by their index in our response letter rather than the index in the main text.

Referee 1:

Dear authors,

First of all, thank you for the detailed and matching response to my comments. I very much enjoyed the direct and bona fide debate. I respond further only to the issues which I believe to be outstanding.

C-1.: I agree that whilst real systems are dirty and may not fit a certain stable probabilistic structure, it doesn't mean there cannot be a useful abstraction. Certainly some of the references given in the response seems to indicate that there can be a degree of consistency in attack behaviour. On a more philosophical point, one may wish to see if there is a need to distinguish network formation laws (e.g. a decentralised complexity) with attacks (which maybe a form of centralised complexity). What would be useful is to perhaps set out 1-2 mechanics for real-world attacks that obey a certain pattern in the SI. This would help to motivate readers and also expand on the impact of the paper.

Our Response: We thank the reviewer for all the insightful comments and we do share the same joy as the reviewer in this interactive process of learning and discussion to improve our manuscript.

We have followed the reviewer's suggestion to add discussion in SI (see S6) on an identified instance of social network attack by Russian Internet Research Agency on twitter networks. We also put pointers in our main discussion to motivate the readers for further reading on relevant topics.

F: I would suggest also pointing out where this approach might not work in the paper (which also relates to some C-1 points). Often knowing its limitations is very helpful for end users and also enables others to follow up on further research. It maybe the case that limitations are better demonstrated with simpler examples (which chimes with another reviewer's comments I believe).

Our Response: We thank the reviewer for this important comment . We have followed the suggestion and added the discussion of limitation of our approach in the section "Discussion and Future work" which is highlighted in red. In addition, we also discussed the limitations and possible improvement as we analyze the results.

For convenience, we reproduce them as below.

There are several key aspects that could be improved by our future work. While the assumptions made in the attack model seem plausible, the real attacks may not follow a consistent statistical pattern as the one described in Equation (2). For instance, the causal structure of the attacking sequence considered in our framework can be more sophisticated by the coordination / interaction (sharing of information) among multiple attackers co-existing in the network. Attackers may not necessarily operate under the same strategy which makes it challenging to construct consistent and accurate models to characterize their behavior. Consequently, it is important to incorporate attack strategies as part of the network inference framework (e.g., either estimating the unknown parameters of the attack model together with the network model in an EM approach or estimate them separately based on additional information when available). Since real networks can change their growth rules and possibly their self-similar structure over time (e.g., co-existence or emergent transition of small-world and multi-fractal

Figure 1: Inference runtime as a function of missing nodes with a network of size 4049 and 1015

scaling observed in complex networks), a generative model that captures all the structural features of interested networks can be difficult to build. Finally, applying the inference framework to large scale networks would require more efficient computational techniques. As detailed in the Methods section, the overall computation complexity of one EM iteration is $O(KS|E_0|)$ where $|E_0|$ is the number of links in the original network, K is the number of samples and S denotes optimization steps. In the worst case, $|E_0|$ is a quadratic function of network size and the number of samples required to identify the network model also grows exponentially. This can be shown in Figure 9 that runtime is dominated by the network size and slowly increases as the $|Z_t|$ grows where $|Z_t|$ is the number of latent nodes. The algorithm is written in Matlab and runs on i7-4790K with 32GB memory where $K = 40000$, $B = 10000$ and $S = 10$.

While extending the inference framework to larger scales requires further work, we also need to be very cautious about the interpretation of the worst-case computational complexity. Firstly, many real networks are sparse, which makes the runtime of proposed algorithm run much faster than the worse-case computational complexity implies. Secondly, the size of many biological networks varies from a few hundreds to a few thousands of nodes, which makes the proposed framework suitable for use and further extension to specific biological investigations. Thirdly, social networks are known to possess small world and scale free properties, as well as rich in the degree of locality (related to occupation, age, or geographic proximity). Also, attackers can hardly grasp the global information about the networks. This means that a targeted attack usually happens to a localized subnetwork (observable part of the network for the attacker) rather than the entire network. Combining these important aspects with more realistic attack strategies and opinion / information diffusion models opens up a rich yet challenging class of network reconstruction and inference problems for the network science research community.

H. I am very happy that the paper is better worded and when adding in my recommendation for setting clear limits for the graph and attack structure boundaries (see C-1 and F), I think it makes a solid and impactful contribution to this body of important literature.

In summary, I recommend an accept (subject to minor changes recommended above).

Our Response: We thank again very much the reviewer for all the improvement suggestions and sincere support.

Referee 2:

C-1: This is a follow up on my first review (I was reviewer 2). I read the reply to my comments carefully. If I am trying to be witty, I would say that the problem with "Catching the invisible" is that the authors want to "have their cake and eat it too".

The authors argue in their reply that their manuscript is providing a first step on a new approach to uncover the presence and impact of hidden, antagonistic networks of agents. This is a good and important goal. However, then authors then write their manuscript as if they have made a much greater advance than they actually achieved. This is my problem!

The paper is written as if their approach is going to make visible what happened in the 2016 US Presidential Elections and the 2016 Brexit Referendum. This is extremely misleading and something that I hope Nature Communications will not condone.

If the authors want to write a serious paper, then they should consider several network models of increasing complexity and several attacking strategies of increasing complexity, and several attempts at guessing the parameters of the attacks. And then, they should write about the conditions under which one can or cannot "catch the invisible".

As is, the manuscript is a misleading exercise whose aim appears to be getting press attention.

Our Response: We are thankful to the reviewer's comment. We recognize that our presented election example is an idealized scenario where our proposed method can help reveal the structural properties of latent network structures under a class of attacks, which are abstractions of real attacks that are far more complicated in terms of their scales, dynamics and statistical consistency. Following the reviewer's suggestions, we have thoroughly rewritten and reintroduced our study on the social network to minimize any possible claims and arguments that might trigger confusion and misinterpretation. For convenience, we have reproduced the relevant discussion here in highlight.

I.Reintroduction of the social network example:

In the second experiment, we use our framework to discover the hidden subnetwork in a simulated removal process that mimics the social network interventions in an abstracted setting. This study is inspired by the recent social network user privacy and information breaches. For instance, automated applications run by data firms or malicious attackers are injected into the social networks. These injected applications become part of the social networks and either (i) act as collectors to gather privacy related user profiles under a camouflaged data acquisition interface (e.g., "This is your digital life" that collects user profiles, which are later used for political purpose) or (ii) launch campaigns to propagate designed information to target social groups. Together with the user nodes, they form an extended network that is usually not fully unveiled. The ultimate challenge is to estimate their structural formation and influence on various social events.

It should be noted that assessing the impact of these automated information disseminators and collectors comprehensively requires a sophisticated integration of network inference framework, opinion diffusion dynamics under various attack strategies and scales, geometry and statistical physics, network science and even psychological profiling and modeling. As one of the key enablers towards a reliable toolset against such information manipulations, we evaluate, from a structural perspective, the inference capability of a properly built framework that admits and exploits the knowledge of the attack and how it can be leveraged to boost the fidelity of recovered network structures.

II.Reintroduced result analysis according to reviewer's suggestion:

Discover the hidden social networks.

In the following experiment, we connect our case study to the social network interventions that are related to recent ever-increasing privacy and information manipulation concerns. We focus on the capability of our framework to retrieve with fidelity the structure of a subnetwork removed under our targeted attack assumption in Equation (2). This targeted removal process mimics the removal of automated applications deployed intentionally to either collect or inject information into social networks. In the extended social network consisting of both such hidden applications and user nodes, such a removal process can be understood as a defensive strategy of the launchers (e.g., data firms) to get minimal exposure to the investigation by pulling the deployed applications offline. Different from the attack in previous experiments, the removal process now obstructs our observations rather than sabotaging the network entities.

Although real social network attacks can be much more sophisticated by involving multiple parties at the same time (as opposed to a coordinated sequence of operations as in Equation (2) and evolving in a statistically inconsistent way (as opposed to a stabilized and consistent stochastic behavior), here we consider an idealized abstraction of a class of real attacks that prioritize the degree centrality. The considered attack model and its variants have been widely adopted as an abstraction of the targeted attacks for the study of robustness, stability, resilience and defensive/attack strategies of networks [26, 30, 38-44] ranging from mathematically constructed complex network to traffic network [45], brain network [46-48], computer network [13] and also social networks [49, 50]. Of particular note is the fact that the lack of global information in a social network attack is common whereas the probability of reaching a particular vertex by following a randomly chosen edge in a graph is proportional to the vertex’s degree [30], making the degree centrality an important factor that contributes to the vulnerability of the nodes even though the attacker has only extremely localized information (e.g., connectivity). Moreover, the study [26] suggests that the choice of α in the Equation (2) can be used to incorporate the intrinsic network vulnerability and external knowledge of the system, which helps the model and its variants become a good abstraction of a wide range of real attacks in complex networks.

Starting out from this motivation, we consider an extended social network with 4049 nodes (including hidden nodes injected for information manipulation, referred as injected nodes, and ordinary user nodes) built from Facebook network dataset [51]. Due to the small-worldness of the social network (see Supplementary Material S2), only a small group of injected nodes is required to make sure all user nodes have at least one injected node as their immediate neighbor (i.e., all users are subject to data security issues and/or manipulated information even without information propagation among them). We define coverage to be the chance of a user node to have an immediate neighboring injected node. Figure 10 visualizes the coverage of injected nodes against their share in the network under different α from -10 to 10 . In this figure, α has a different meaning and $\mathcal{A}_\alpha(d_i)$ now is a proxy of the likelihood of an injected node of degree d_i being the highest connected node in the network. The higher α is, the larger portion of the highest connected nodes will be represented by the injected nodes and the bigger coverage they will have. Figure 10 suggests that 48.6% of population have at least one neighboring injected node when the injected nodes account for only 1% of total nodes with $\alpha = 1$. The coverage goes up to 98.44% when injected nodes account for 15% of the network as shown in Figure 10. This suggests that a full-scale information manipulation/collection requires only a small injection of designed agents (i.e., disseminators/collectors) into the network and these agents do not have to be significantly more connected than an average node.

Motivated by this observation, we simulate the removal process by setting $\alpha = 1$ and vary the share of injected nodes from 5% to 45%. Baseline method and our framework are applied to recover the original extended social network. Similarly, ROC-AUC and PR-AUC are used as metrics for quantifying the inference capability and shown in Figure 11.(a) and (c). Resonating with our previous experiments, the ROC-AUC and PR-AUC scores of our proposed inference

Figure 2: Coverage as a function of α and share of injected nodes in total network of 4049 nodes

framework are significantly improved over that of the baseline, suggesting a boost in capability to infer the missing network more accurately. This is further corroborated by assessing the structural similarity of networks generated by the retrieved network models. Similarly, we estimate the Kolmogorov-Smirnov (KS) distance e_{KS} between the empirical degree distribution of the original network $F^*(x)$ and networks generated by both methods $F(x)$. The results are averaged over 1000 network instances and reported in Figure 11.(c). In addition, we also report the log-likelihood (LL) in Figure 11.(d) as a global metric for goodness-of-fit to compare the model identified by both methods. Even though the absolute value of LL strongly varies as a function of a particular model choice for the network, the relative difference given the fixed model provides a good performance comparison between different identification techniques. As expected, Figure 11.(b) and (d) suggest that our proposed method retrieves a model that is more globally consistent with the true one with smaller e_{KS} and larger LL values compared to the baseline.

The statistics of both the intervention process and the complex network structure play a crucial role in these observations. First, in small-world networks, the hub nodes account for a small fraction of the network. Lower degree nodes are unaffected by hub-prioritized interventions. The baseline method ignores the influence of the intervention and therefore is biased by the observed part towards the retrieval of a model that explains better a network without the hub nodes. As demonstrated by our studies, the baseline method has poor performance on inferring the missing network. Second, due to the time-varying nature of the interventions, the hub-prioritized interventions induce a random sampling behavior after the removal of hub nodes. This behavior change can be demonstrated by the small variance of the degree distribution, reshaped by the conducted intervention (see Transitional behavior of interventions in Supplementary Material). Consequently, the performance of baseline and proposed methods exhibit a plateau since a small-world network is robust against random removals. We present the investigation of small-world-ness of all networks considered in our work in Supplemen-

Figure 3: **Evaluation of the capability to recover the Facebook social network ($\alpha = 1$).** (a, c) Compare the capability to infer the missing network via AUC scores. (b) Goodness-of-the-fit comparison as reported by Kolmogorov-Smirnov distance between the true degree distribution and the one retrieved by baseline and proposed methods. (d) Quantification of the capability of both methods to recover the global property of the example social network via log-likelihood.

tary Material (S2).

Last but not least, we report the estimated number of user nodes (later referred as "affected users") with at least one injected node as their immediate neighbor. Without considering the opinion diffusion dynamics, this measurement serves as an upper bound on the number of users being exposed to designed information or personal data breaches. To consider a more realistic setting, this assessment should also incorporate the propagation of information among users, which is left as an important extension in our future work. Varying the share of injected nodes in the extended social network from 1% to 15%, Figure 12 visualizes the estimation averaged over 5000 network instances drawn from both models retrieved by baseline and proposed methods. As expected, the baseline underestimates affected users as it did not exploit the knowledge of the targeted removal process. More interestingly, when compared to Figure 10, we found that the curve corresponding to the estimated votes by the baseline is almost identical to the coverage curve obtained under a random intervention (i.e., the degree of an injected node being statistically the same as a randomly chosen node in the original network without injected nodes). This suggests again that the baseline method works only if the intervention is purely randomized and easily fails when this assumption does not hold.

We have also added a detailed discussion on the limitation of our proposed framework to clearly state its applicability and scalability. Please kindly see the text highlighted in red on Page 4-5, Page 6-7 and Page 8-9 and we have also reproduced here in highlight.

Discussion of the limitation of proposed framework:

There are several key aspects that could be improved by our future work. While the assumptions made in the attack model seem plausible, the real attacks may not follow a con-

Figure 4: Comparison of capability to estimate affected users ($\alpha = 1$)

sistent statistical pattern as the one described in Equation (2). For instance, the causal structure of the attacking sequence considered in our framework can be more sophisticated by the coordination / interaction (sharing of information) among multiple attackers co-existing in the network. Attackers may not necessarily operate under the same strategy which makes it challenging to construct consistent and accurate models to characterize their behavior. Consequently, it is important to incorporate attack strategies as part of the network inference framework (e.g., either estimating the unknown parameters of the attack model together with the network model in an EM approach or estimate them separately based on additional information when available). Since real networks can change their growth rules and possibly their self-similar structure over time (e.g., co-existence or emergent transition of small-world and multi-fractal scaling observed in complex networks [55,56]), a generative model that captures all the structural features of interested networks can be difficult to build. Finally, applying the inference framework to large scale networks would require more efficient computational techniques. As detailed in the Methods section, the overall computation complexity of one EM iteration is $O(KS|E_0|)$ where $|E_0|$ is the number of links in the original network, K is the number of samples and S denotes optimization steps. In the worst case, $|E_0|$ is a quadratic function of network size and the number of samples required to identify the network model also grows exponentially. This can be shown in Figure 9 that runtime is dominated by the network size and slowly increases as the $|Z_t|$ grows where $|Z_t|$ is the number of latent nodes. The algorithm is written in Matlab and runs on i7-4790K with 32GB memory where $K = 40000$, $B = 10000$ and $S = 10$.

While extending the inference framework to larger scales requires further work, we also need to be very cautious about the interpretation of the worst-case computational complexity. Firstly, many real networks are sparse, which makes the runtime of proposed algorithm run much faster than the worst-case computational complexity implies. Secondly, the size of many biological networks varies from a few hundreds to a few thousands of nodes, which makes the proposed framework suitable for use and further extension to specific biological investigations. Thirdly, social networks are known to possess small world and scale free properties, as well as rich in the degree of locality (related to occupation, age, or geographic proximity). Also, attackers can hardly grasp the global information about the networks. This means that a targeted attack usually happens to a localized subnetwork (observable part of the network for the attacker) rather than the entire network. Combining these important aspects with

Figure 5: Inference runtime as a function of missing nodes with a network of size 4049 and 1015

more realistic attack strategies and opinion / information diffusion models opens up a rich yet challenging class of network reconstruction and inference problems for the network science research community.

Referee 3:

C-1: The authors have tried to address all my issues and to improve the presentation of their work. In particular, adding Fig. 1 in the main text with an introductory example made the article clearer. I also appreciate the addition of the part about the algorithmic complexity in the Supplementary Material.

Our Response: We are very thankful to the reviewer's comments. All the suggestions and discussion helped us improve our work. We also hope to address all the further comments as best as we can.

C-2: I think that the novel method proposed in this work represents an important advance in the field of network reconstruction and could also be of interest for a broader audience dealing with complex network and may be suited for publication in Nature Communications. However, I think the current version of the manuscript should still be improved. In particular, the authors could do a better job of taking into account the issues raised by the referees and integrating their answers in the manuscript. Firstly, it is difficult to understand what changes were done to the new version of the manuscript since they do not indicate it clearly. They could have provided a version of the article with the changes highlighted and reported all the changes and their location in the answer to the referees letter (they did it for some changes but not consistently). Even if this is, in a sense, a new submission (since it was transferred from Nature), this would have made their work much clearer.

Our Response: We are very thankful to the reviewer for guiding us to better present our changes between revision by highlighting and integration. We have followed the suggestion and incorporated the key discussions on relevant topics to the manuscript. In addition, we have prepared two versions of our revised manuscript (one with changes merged and another one with all the changes highlighted) for better readability. To help the reviewer easily identify the new revisions, we also reproduce the relevant changes in accordance to each individual comment.

About the example of the election manipulation

C-3: I appreciate the clearer introduction of this example on page 6. However I still think that this example is misleading.

I agree that, in a very idealized model of election manipulation, their method may be used to "discover the hidden subnetwork of attackers who try to manipulate people's opinion with a social network", but I do not see how it could be used to measure the real influence of the attackers on the vote in a real world scenario. While they addressed some of my remarks concerning this point in the answer to the referees letter, they did not include the remarks about the limitations of their method in the article.

If they persist in wanting to use this case as an example of application of their method, they need to clearly explain which questions it can answer and which it cannot. They need to stipulate that their simulation is highly idealized and clarify the limitations of this example in order to not mislead readers into thinking that their approach solves the problem of opinion manipulation as this is a much more complex problem than what they simulate.

Our Response: We are very thankful to the reviewer's comment on this example. We recognize that our presented election example is an idealized scenario where our proposed method can help reveal the structural properties of latent network structures under a class of attacks, which are abstractions of real attacks that are far more complicated in terms of their scales,

dynamics and statistical consistency. Following the suggestion from the reviewer, we have reintroduced our case study on the social network and added the detailed discussion on the current limitations of our framework for future extension and improvement. To minimize any possible claims and arguments that might trigger confusion and misinterpretation, we have followed the suggestion of the reviewers to clearly state the applicability and scalability of our framework. Please kindly see highlighted text on Page 4-9. For convenience, we reproduced them here comprehensively. We will refer to them more specifically in our response to the following comments.

I.Reintroduction of the social network example:

In the second experiment, we use our framework to discover the hidden subnetwork in a simulated removal process that mimics the social network interventions in an abstracted setting. This study is inspired by the recent social network user privacy and information breaches. For instance, automated applications run by data firms or malicious attackers are injected into the social networks. These injected applications become part of the social networks and either (i) act as collectors to gather privacy related user profiles under a camouflaged data acquisition interface (e.g., "This is your digital life" that collects user profiles, which are later used for political purpose) or (ii) launch campaigns to propagate designed information to target social groups. Together with the user nodes, they form an extended network that is usually not fully unveiled. The ultimate challenge is to estimate their structural formation and influence on various social events.

It should be noted that assessing the impact of these automated information disseminators and collectors comprehensively requires a sophisticated integration of network inference framework, opinion diffusion dynamics under various attack strategies and scales, geometry and statistical physics, network science and even psychological profiling and modeling. As one of the key enablers towards a reliable toolset against such information manipulations, we evaluate, from a structural perspective, the inference capability of a properly built framework that admits and exploits the knowledge of the attack and how it can be leveraged to boost the fidelity of recovered network structures.

II.Reintroduced result analysis according to reviewer's suggestion:

Discover the hidden social networks.

In the following experiment, we connect our case study to the social network interventions that are related to recent ever-increasing privacy and information manipulation concerns. We focus on the capability of our framework to retrieve with fidelity the structure of a subnetwork removed under our targeted attack assumption in Equation (2). This targeted removal process mimics the removal of automated applications deployed intentionally to either collect or inject information into social networks. In the extended social network consisting of both such hidden applications and user nodes, such a removal process can be understood as a defensive strategy of the launchers (e.g., data firms) to get minimal exposure to the investigation by pulling the deployed applications offline. Different from the attack in previous experiments, the removal process now obstructs our observations rather than sabotaging the network entities.

Although real social network attacks can be much more sophisticated by involving multiple parties at the same time (as opposed to a coordinated sequence of operations as in Equation (2) and evolving in a statistically inconsistent way (as opposed to a stabilized and consistent stochastic behavior), here we consider an idealized abstraction of a class of real attacks that prioritize the degree centrality. The considered attack model and its variants have been widely adopted as an abstraction of the targeted attacks for the study of robustness, stability, resilience and defensive/attack strategies of networks [26, 30, 38-44] ranging from mathematically constructed complex network to traffic network [45], brain network [46-48], computer network

[13] and also social networks [49, 50]. Of particular note is the fact that the lack of global information in a social network attack is common whereas the probability of reaching a particular vertex by following a randomly chosen edge in a graph is proportional to the vertex’s degree [30], making the degree centrality an important factor that contributes to the vulnerability of the nodes even though the attacker has only extremely localized information (e.g., connectivity). Moreover, the study [26] suggests that the choice of α in the Equation (2) can be used to incorporate the intrinsic network vulnerability and external knowledge of the system, which helps the model and its variants become a good abstraction of a wide range of real attacks in complex networks.

Starting out from this motivation, we consider an extended social network with 4049 nodes (including hidden nodes injected for information manipulation, referred as injected nodes, and ordinary user nodes) built from Facebook network dataset [51]. Due to the small-worldness of the social network (see Supplementary Material S2), only a small group of injected nodes is required to make sure all user nodes have at least one injected node as their immediate neighbor (i.e., all users are subject to data security issues and/or manipulated information even without information propagation among them). We define coverage to be the chance of a user node to have an immediate neighboring injected node. Figure 10 visualizes the coverage of injected nodes against their share in the network under different α from -10 to 10 . In this figure, α has a different meaning and $\mathcal{A}_\alpha(d_i)$ now is a proxy of the likelihood of an injected node of degree d_i being the highest connected node in the network. The higher α is, the larger portion of the highest connected nodes will be represented by the injected nodes and the bigger coverage they will have. Figure 10 suggests that 48.6% of population have at least one neighboring injected node when the injected nodes account for only 1% of total nodes with $\alpha = 1$. The coverage goes up to 98.44% when injected nodes account for 15% of the network as shown in Figure 10. This suggests that a full-scale information manipulation/collection requires only a small injection of designed agents (i.e., disseminators/collectors) into the network and these agents do not have to be significantly more connected than an average node.

Motivated by this observation, we simulate the removal process by setting $\alpha = 1$ and vary the share of injected nodes from 5% to 45%. Baseline method and our framework are applied to recover the original extended social network. Similarly, ROC-AUC and PR-AUC are used as metrics for quantifying the inference capability and shown in Figure 11.(a) and (c). Resonating with our previous experiments, the ROC-AUC and PR-AUC scores of our proposed inference framework are significantly improved over that of the baseline, suggesting a boost in capability to infer the missing network more accurately. This is further corroborated by assessing the structural similarity of networks generated by the retrieved network models. Similarly, we estimate the Kolmogorov-Smirnov (KS) distance e_{KS} between the empirical degree distribution of the original network $F^*(x)$ and networks generated by both methods $F(x)$. The results are averaged over 1000 network instances and reported in Figure 11.(c). In addition, we also report the log-likelihood (LL) in Figure 11.(d) as a global metric for goodness-of-fit to compare the model identified by both methods. Even though the absolute value of LL strongly varies as a function of a particular model choice for the network, the relative difference given the fixed model provides a good performance comparison between different identification techniques. As expected, Figure 11.(b) and (d) suggest that our proposed method retrieves a model that is more globally consistent with the true one with smaller e_{KS} and larger LL values compared to the baseline.

The statistics of both the intervention process and the complex network structure play a crucial role in these observations. First, in small-world networks, the hub nodes account for a small fraction of the network. Lower degree nodes are unaffected by hub-prioritized interventions. The baseline method ignores the influence of the intervention and therefore is biased by the observed part towards the retrieval of a model that explains better a network without the hub nodes. As demonstrated by our studies, the baseline method has poor performance

Figure 6: Coverage as a function of α and share of injected nodes in total network of 4049 nodes

on inferring the missing network. Second, due to the time-varying nature of the interventions, the hub-prioritized interventions induce a random sampling behavior after the removal of hub nodes. This behavior change can be demonstrated by the small variance of the degree distribution, reshaped by the conducted intervention (see Transitional behavior of interventions in Supplementary Material). Consequently, the performance of baseline and proposed methods exhibit a plateau since a small-world network is robust against random removals. We present the investigation of small-world-ness of all networks considered in our work in Supplementary Material (S3).

Last but not least, we report the estimated number of user nodes (later referred as "affected users") with at least one injected node as their immediate neighbor. Without considering the opinion diffusion dynamics, this measurement serves as an upper bound on the number of users being exposed to designed information or personal data breaches. To consider a more realistic setting, this assessment should also incorporate the propagation of information among users, which is left as an important extension in our future work. Varying the share of injected nodes in the extended social network from 1% to 15%, Figure 12 visualizes the estimation averaged over 5000 network instances drawn from both models retrieved by baseline and proposed methods. As expected, the baseline underestimates affected users as it did not exploit the knowledge of the targeted removal process. More interestingly, when compared to Figure 10, we found that the curve corresponding to the estimated votes by the baseline is almost identical to the coverage curve obtained under a random intervention (i.e., the degree of an injected node being statistically the same as a randomly chosen node in the original network without injected nodes). This suggests again that the baseline method works only if the intervention is purely randomized and easily fails when this assumption does not hold.

III. Discussion of the limitation of proposed framework:

There are several key aspects that could be improved by our future work. While the as-

Figure 7: **Evaluation of the capability to recover the Facebook social network ($\alpha = 1$).** (a, c) Compare the capability to infer the missing network via AUC scores. (b) Goodness-of-the-fit comparison as reported by Kolmogorov-Smirnov distance between the true degree distribution and the one retrieved by baseline and proposed methods. (d) Quantification of the capability of both methods to recover the global property of the example social network via log-likelihood.

assumptions made in the attack model seem plausible, the real attacks may not follow a consistent statistical pattern as the one described in Equation (2). For instance, the causal structure of the attacking sequence considered in our framework can be more sophisticated by the coordination / interaction (sharing of information) among multiple attackers co-existing in the network. Attackers may not necessarily operate under the same strategy which makes it challenging to construct consistent and accurate models to characterize their behavior. Consequently, it is important to incorporate attack strategies as part of the network inference framework (e.g., either estimating the unknown parameters of the attack model together with the network model in an EM approach or estimate them separately based on additional information when available). Since real networks can change their growth rules and possibly their self-similar structure over time (e.g., co-existence or emergent transition of small-world and multi-fractal scaling observed in complex networks [55,56]), a generative model that captures all the structural features of interested networks can be difficult to build. Finally, applying the inference framework to large scale networks would require more efficient computational techniques. As detailed in the Methods section, the overall computation complexity of one EM iteration is $O(KS|E_0|)$ where $|E_0|$ is the number of links in the original network, K is the number of samples and S denotes optimization steps. In the worst case, $|E_0|$ is a quadratic function of network size and the number of samples required to identify the network model also grows exponentially. This can be shown in Figure 9 that runtime is dominated by the network size and slowly increases as the $|Z_t|$ grows where $|Z_t|$ is the number of latent nodes. The algorithm is written in Matlab and runs on i7-4790K with 32GB memory where $K = 40000$, $B = 10000$ and $S = 10$.

While extending the inference framework to larger scales requires further work, we also

Figure 8: Comparison of capability to estimate affected users ($\alpha = 1$)

Figure 9: Inference runtime as a function of missing nodes with a network of size 4049 and 1015

need to be very cautious about the interpretation of the worst-case computational complexity. Firstly, many real networks are sparse, which makes the runtime of proposed algorithm run much faster than the worst-case computational complexity implies. Secondly, the size of many biological networks varies from a few hundreds to a few thousands of nodes, which makes the proposed framework suitable for use and further extension to specific biological investigations. Thirdly, social networks are known to possess small world and scale free properties, as well as rich in the degree of locality (related to occupation, age, or geographic proximity). Also, attackers can hardly grasp the global information about the networks. This means that a targeted attack usually happens to a localized subnetwork (observable part of the network for the attacker) rather than the entire network. Combining these important aspects with more realistic attack strategies and opinion / information diffusion models opens up a rich yet challenging class of network reconstruction and inference problems for the network science research community.

C-4: They use the case of Cambridge Analytica to justify their example, saying "We simulate a similar opinion manipulation attack as the reported case of Cambridge Analytica on Face-

book users with the difference being that we assume there exists a multitude of running agents that try to influence voting decisions.” However, the case of Cambridge Analytica is not very similar to their scenario. The app used by Cambridge Analytica was only used to gather information on people using it (and on their friends) in order to build psychological profiles.

This was then used to target users based on their personality traits with digital ads and fundraising appeals (see <https://www.nytimes.com/2018/03/17/us/politics/cambridge-analytica-trump-campaign.html>). Comparing their simulation to the case of Cambridge Analytical shows that a realistic simulation should in fact include a targeting that depends on psychological profiles and influence outside of just the neighbors of a users (ads).

I think this is what the author mean in the answer to referee letter, when they say “Another important note here is that there is no influence propagation or opinion/rumor spreading in this type of social network interventions. They can be triggered by the actions performed by the data firms after the attack but they are not participating in it through the “agent apps” by themselves.” But this is not precised in the main text.

Their scenario is closer to the case of opinion manipulation using bots which I don’t think we have proof that Cambridge Analytical used. If I understand correctly, this is what they mean by automated apps in the answer to referee letter. They should clarify this point in the manuscript.

For an empirical investigation of opinion manipulation with bots, see for example : Ferrara, Emilio, et al. “The rise of social bots.” *Communications of the ACM* 59.7 (2016): 96-104 and Bessi, Alessandro, and Emilio Ferrara. “Social bots distort the 2016 US Presidential election online discussion.” (2016).

Our Response: We are very thankful to the reviewer for helping us clarify our claims with respect to the automated apps that try to influence the voting decisions. As the reviewer mentioned, we were motivated by the attack scenario where political campaigns are deployed by means of automated apps (e.g., news bots) running on top of social media platforms like Facebook or Twitter, based on the collected psychological profiles of users.

To support a campaign that targets millions of users on the social network (<https://www.theguardian.com/news/2018/may/06/cambridge-analytica-how-turn-clicks-into-votes-christopher-wylie>), we believe that data firms like Cambridge Analytica have to take advantage of automated deployment/broadcasting algorithms and utilities provided by social media platforms to build user profiles and inject designed information based on that. This is also corroborated by a recent study [R1] on the electoral manipulation that is using trolls (malicious accounts created for the purpose of manipulation) and bots (automated accounts) to spread misinformation and politically biased information. Note that these algorithms are not only the social bots as in the reference but also built-in services of social media platforms that are being utilized in a wrong way (e.g., Reveal: <https://revealbot.com/>; Zalster, <https://zalster.com/>).

[R1] Badawy, Adam, Emilio Ferrara, and Kristina Lerman. “Analyzing the Digital Traces of Political Manipulation: The 2016 Russian Interference Twitter Campaign.” arXiv preprint arXiv:1802.04291 (2018).

As mentioned in our response to the previous comment, we have followed the suggestion of the reviewer to make our discussion of this precise with a clear scope to minimize confusion. We have also included the recommended work in our reference. Please kindly see highlighted text on Page 4-9 for a full view of the change. Specifically, please see *I.Reintroduction of the social network example:* and *III.Discussion of the limitation of proposed framework:* of our response to C-3.

C-5: Their explanation of the simulation could also be improved. For example, they speak about “the fraction of population being biased” (also in Figure 5) without clearly defining

what they consider a biased user (is it the neighbors of the planted influencers?).

What is really unclear for me is how they can measure the “number of biased vote” (Figure 7) and at same time say that “there is no influence propagation or opinion/rumor spreading in this type of social network interventions”. If they measure the number of biased vote, they assume that the opinion of the voters was influenced. Otherwise, what they measure may be better describe as the number agent passively collecting data? or the number of user potentially influenced?

But in this case, they need to specify that we have no idea how successful the opinion manipulation is, which means that their result is of little practical use. This sentence: “Last but not least, we report the estimated votes potentially influenced by the agents who directly interacted with people. This represents a lower bound of biased votes without including the magnifying effect of opinion propagation.” is also particularly misleading. “The number of votes potentially influenced” is not equivalent to the number of “biased vote”, since we don’t know how successful the opinion manipulation is. A potentially influenced vote is not necessarily a biased vote. Not everybody will change their opinion when they are in contact with an opinion manipulator. So the number is in fact an upper bond on the number of biased votes (without taking into account the possibility of opinion propagating farther than to immediate neighbors). I would appreciate if the authors could clarify this point in the main text.

Our Response: We thank the reviewer for the suggestion. We have thoroughly reintroduced our social network study to reflect our efforts to apply the suggested changes. More specifically, we presented our analysis of the results with full awareness of the fact that accurate assessment of bias propagation would require an integration of network inference and opinion diffusion dynamics in a more sophisticated setting, which serves as a key extension of our current work in the future (e.g., one possible extension to consider is to adopt the label propagation method similar to the one used in [R1] to employ the subnet of attackers as seeds and propagate the labels or ideology). We have highlighted our changes in red in the reproduced result analysis below (II.Reintroduced result analysis according to reviewer’s suggestion).

With the awareness of the future challenges to improve our method, we would like to also emphasize that opinion dynamics methods that try to understand the influence of social network attackers (e.g., trolls) rely on the knowledge of full network structure. Estimating the influences in a partially observed network by opinion diffusion dynamics requires first to either promote its observability (by enhanced sensing) or make reliable inference on its latent subnet which our proposed framework tries to address. In other words, our work tries to serve as the very first step to solve the challenge and we are looking forward to its integration with other methods towards building a reliable estimation tool for complex network attacks.

As a further validation of proposed framework and the attack model adopted in our case studies, we added a study in our Supplementary Material on the retweet network formed by the identified trolls of Russia Internet Research Agency that tries to compromise the integrity of US 2016 election and reproduced here for the convenience of the reviewer.

The dataset used in the analysis can be downloaded via:

https://about.twitter.com/en_us/values/elections-integrity.html

The list of twitter handles believed to be Russian trolls can be found:

<https://www.recode.net/2017/11/2/16598312/russia-twitter-trump-twitter-deactivated-handle-list>

The processing script is available also on github:

https://github.com/urashima9616/NetworkReconstruction/NetworkReconstruction/twitterprocessing/data_process.py

For convenience, we have highlighted our changes in red in the following reproduced result analysis.

S6: Structural properties study of a real-world Russian Twitter attack by trolls in US 2016 election

To corroborate our assumption that degree centrality plays a key role in the attack and validate the adoption of the attack model class in our case study by a real social network intervention instance, we are considering the on-going investigation by Twitter and US congress on the electoral intervention conducted by Internet Research Agency (IRA) of Russia. Both Twitter and US congress have confirmed the use of trolls (automated malicious bots) by IRA and identified the twitter accounts that helped propagate the manipulated information. Specifically, we measured the degree centrality of trolls identified by US congress as compared to the rest of relevant twitter accounts in a collected tweets dataset.

The election integrity dataset we use is publicly released by Twitter and updated as the investigation proceeds. This dataset consists of 1.8 million Russia-intervention related tweets that involve 71938 users, who formed a retweet network of 703467 links. Among these users, we have identified 100 accounts that are confirmed by US congress as malicious trolls injected by IRA, who is believed to be the main driver of the social media campaign against the US election integrity.

To understand whether these confirmed trolls formed hubs of manipulated information (as assumed by the model in our case study of the social network) in this retweet network, we have measured the out degree of both the identified trolls and the rest of nodes that interacted with trolls (referred as spreaders). We have performed the two-sample t-test with significance level of 0.95 and KS test to compare their degree distributions. The results are summarized in Table 2. Combining the results from both tests, several observations are due.

1. The two degree distributions are significantly different. This corroborates our argument that the distribution of the latent network formed by the injected trolls are not necessarily similar to the observed one. Inference of latent network without the relevant knowledge of the attack might result in large structural deviation.

2. The mean degree of the trolls is much higher (655) than that of the spreaders (9.7788). This supports our assumption that social network attackers are injected in a way to maximize the information spreading by forming hubs with much higher degrees (i.e., significantly higher volume of posts and retweets).

These two observations in this real social network attack suggest that (i) the knowledge about the attack is important to infer the latent network with good fidelity and (ii) attack model based on degree centrality can potentially serve as a reasonable abstraction of their behavior and capture the structural properties of subnetworks formed by these attackers.

Statistics	Trolls	Spreaders
K-S test p-value	$3.3191 * 10^{-59}$	
t-test p-value	$3.2240 * 10^{-234}$	
mean out degree	655	9.7788

Table 1: Comparison of degree centrality statistics of injected Russian Trolls and spreaders in a retweet network

II.Reintroduced result analysis according to reviewer’s suggestion:

Discover the hidden social networks.

In the following experiment, we connect our case study to the social network interventions that are related to recent ever-increasing privacy and information manipulation concerns. We

focus on the capability of our framework to retrieve with fidelity the structure of a subnetwork removed under our targeted attack assumption in Equation (2). This targeted removal process mimics the removal of automated applications deployed intentionally to either collect or inject information into social networks. In the extended social network consisting of both such hidden applications and user nodes, such a removal process can be understood as a defensive strategy of the launchers (e.g., data firms) to get minimal exposure to the investigation by pulling the deployed applications offline. Different from the attack in previous experiments, the removal process now obstructs our observations rather than sabotaging the network entities.

Although real social network attacks can be much more sophisticated by involving multiple parties at the same time (as opposed to a coordinated sequence of operations as in Equation (2) and evolving in a statistically inconsistent way (as opposed to a stabilized and consistent stochastic behavior), here we consider an idealized abstraction of a class of real attacks that prioritize the degree centrality. The considered attack model and its variants have been widely adopted as an abstraction of the targeted attacks for the study of robustness, stability, resilience and defensive/attack strategies of networks [26, 30, 38-44] ranging from mathematically constructed complex network to traffic network [45], brain network [46-48], computer network [13] and also social networks [49, 50]. Of particular note is the fact that the lack of global information in a social network attack is common whereas the probability of reaching a particular vertex by following a randomly chosen edge in a graph is proportional to the vertex’s degree [30], making the degree centrality an important factor that contributes to the vulnerability of the nodes even though the attacker has only extremely localized information (e.g., connectivity). Moreover, the study [26] suggests that the choice of α in the Equation (2) can be used to incorporate the intrinsic network vulnerability and external knowledge of the system, which helps the model and its variants become a good abstraction of a wide range of real attacks in complex networks.

Starting out from this motivation, we consider an extended social network with 4049 nodes (including hidden nodes injected for information manipulation, referred as injected nodes, and ordinary user nodes) built from Facebook network dataset [51]. Due to the small-worldness of the social network (see Supplementary Material S2), only a small group of injected nodes is required to make sure all user nodes have at least one injected node as their immediate neighbor (i.e., all users are subject to data security issues and/or manipulated information even without information propagation among them). We define coverage to be the chance of a user node to have an immediate neighboring injected node. Figure 10 visualizes the coverage of injected nodes against their share in the network under different α from -10 to 10 . In this figure, α has a different meaning and $\mathcal{A}_\alpha(d_i)$ now is a proxy of the likelihood of an injected node of degree d_i being the highest connected node in the network. The higher α is, the larger portion of the highest connected nodes will be represented by the injected nodes and the bigger coverage they will have. Figure 10 suggests that 48.6% of population have at least one neighboring injected node when the injected nodes account for only 1% of total nodes with $\alpha = 1$. The coverage goes up to 98.44% when injected nodes account for 15% of the network as shown in Figure 10. This suggests that a full-scale information manipulation/collection requires only a small injection of designed agents (i.e., disseminators/collectors) into the network and these agents do not have to be significantly more connected than an average node.

Motivated by this observation, we simulate the removal process by setting $\alpha = 1$ and vary the share of injected nodes from 5% to 45%. Baseline method and our framework are applied to recover the original extended social network. Similarly, ROC-AUC and PR-AUC are used as metrics for quantifying the inference capability and shown in Figure 11.(a) and (c). Resonating with our previous experiments, the ROC-AUC and PR-AUC scores of our proposed inference framework are significantly improved over that of the baseline, suggesting a boost in capability to infer the missing network more accurately. This is further corroborated by assessing

Figure 10: Coverage as a function of α and share of injected nodes in total network of 4049 nodes

the structural similarity of networks generated by the retrieved network models. Similarly, we estimate the Kolmogorov-Smirnov (KS) distance e_{KS} between the empirical degree distribution of the original network $F^*(x)$ and networks generated by both methods $F(x)$. The results are averaged over 1000 network instances and reported in Figure 11.(c). In addition, we also report the log-likelihood (LL) in Figure 11.(d) as a global metric for goodness-of-fit to compare the model identified by both methods. Even though the absolute value of LL strongly varies as a function of a particular model choice for the network, the relative difference given the fixed model provides a good performance comparison between different identification techniques. As expected, Figure 11.(b) and (d) suggest that our proposed method retrieves a model that is more globally consistent with the true one with smaller e_{KS} and larger LL values compared to the baseline.

The statistics of both the intervention process and the complex network structure play a crucial role in these observations. First, in small-world networks, the hub nodes account for a small fraction of the network. Lower degree nodes are unaffected by hub-prioritized interventions. The baseline method ignores the influence of the intervention and therefore is biased by the observed part towards the retrieval of a model that explains better a network without the hub nodes. As demonstrated by our studies, the baseline method has poor performance on inferring the missing network. Second, due to the time-varying nature of the interventions, the hub-prioritized interventions induce a random sampling behavior after the removal of hub nodes. This behavior change can be demonstrated by the small variance of the degree distribution, reshaped by the conducted intervention (see Transitional behavior of interventions in Supplementary Material). Consequently, the performance of baseline and proposed methods exhibit a plateau since a small-world network is robust against random removals. We present the investigation of small-world-ness of all networks considered in our work in Supplementary Material (S3).

Figure 11: **Evaluation of the capability to recover the Facebook social network ($\alpha = 1$).** (a, c) Compare the capability to infer the missing network via AUC scores. (b) Goodness-of-fit comparison as reported by Kolmogorov-Smirnov distance between the true degree distribution and the one retrieved by baseline and proposed methods. (d) Quantification of the capability of both methods to recover the global property of the example social network via log-likelihood.

Last but not least, we report the estimated number of user nodes (later referred as “affected users”) with at least one injected node as their immediate neighbor. Without considering the opinion diffusion dynamics, this measurement serves as an upper bound on the number of users being exposed to designed information or personal data breaches. To consider a more realistic setting, this assessment should also incorporate the propagation of information among users, which is left as an important extension in our future work. Varying the share of injected nodes in the extended social network from 1% to 15%, Figure 12 visualizes the estimation averaged over 5000 network instances drawn from both models retrieved by baseline and proposed methods. As expected, the baseline underestimates affected users as it did not exploit the knowledge of the targeted removal process. More interestingly, when compared to Figure 10, we found that the curve corresponding to the estimated votes by the baseline is almost identical to the coverage curve obtained under a random intervention (i.e., the degree of an injected node being statistically the same as a randomly chosen node in the original network without injected nodes). This suggests again that the baseline method works only if the intervention is purely randomized and easily fails when this assumption does not hold.

C-6: Also in the text we read “With the agents being 1% of the population when $\alpha == 1$, 61.28% of population is already covered”, but in Fig. 5 the proportion seems to be rather just below 50% for $\alpha = 1$ and a share of spreaders of 0.01. Unless I am misreading the figure.

The value of alpha used in Figs 6 and 7 should be mentioned in the caption (I think it’s 1).

Our Response: We thank the reviewer for mentioning this typo we made in the figure due

Figure 12: Comparison of capability to estimate affected users ($\alpha = 1$)

to an earlier version of figures. The coverage measured at Figure 6.(c) should be corrected to be consistent with Figure 6.(b) to be 48.6% for $\alpha = 1$ and a share of spreaders of 0.01 and 80.1% for a share of spreaders of 0.05. We have also applied the changes to the main text when they are discussed (Page 7, paragraph 2 highlighted in red). Please see the excerpt of our response *II.Reintroduced result analysis according to reviewer's suggestion:* to C-3 with respect to our changes:

Starting out from this motivation, we consider an extended social network with 4049 nodes (including hidden nodes injected for information manipulation, referred as injected nodes, and ordinary user nodes) built from Facebook network dataset [51]. Due to the small-worldness of the social network (see Supplementary Material S2), only a small group of injected nodes is required to make sure all user nodes have at least one injected node as their immediate neighbor (i.e., all users are subject to data security issues and/or manipulated information even without information propagation among them). We define coverage to be the chance of a user node to have an immediate neighboring injected node. Figure 10 visualizes the coverage of injected nodes against their share in the network under different α from -10 to 10 . In this figure, α has a different meaning and $\mathcal{A}_\alpha(d_i)$ now is a proxy of the likelihood of an injected node of degree d_i being the highest connected node in the network. The higher α is, the larger portion of the highest connected nodes will be represented by the injected nodes and the bigger coverage they will have. Figure 10 suggests that 48.6% of population have at least one neighboring injected node when the injected nodes account for only 1% of total nodes with $\alpha = 1$. The coverage goes up to 98.44% when injected nodes account for 15% of the network as shown in Figure 10. This suggests that a full-scale information manipulation/collection requires only a small injection of designed agents (i.e., disseminators/collectors) into the network and these agents do not have to be significantly more connected than an average node.

Concerning the issue of algorithmic complexity and running time:

C-7: I appreciate the answer of the referee and the supplementary material they added. However, I don't see any reference to this discussion in the main text (apart for a short sentence at the very end of the manuscript).

I am not blaming the authors for not running their algorithms on more powerful machines, but I think they should mention in the main text up to which order magnitude of network size their framework could reasonably be applied. This has direct implication on the type of

problems that can be solved with their framework.

Our Response: We thank the reviewer very much for his comment. We have added the discussion together with analysis of the limitations of our framework. They are highlighted in red on Page 8-9 and we have reproduced it in our response *III. Discussion of the limitation of proposed framework* to C-3.

For questions C-6, C-7, C-8 and C-9:

C-8: I thank the authors for their answer, but again, a small mention of these points in the main text would improve the manuscript for readers that may have the same questions.

Our Response: We thank the reviewer very much for his comment. We followed carefully all reviewers suggestions and strived to make changes to the best of our ability. To avoid being verbose, we would like to invite the reviewer kindly visit our response to C-3 to see the reproduced changes based on the suggestions. A comprehensive view of these changes can also be found in highlighted text on Page 4-9 of the manuscript.

C-9: There still are some inconsistencies in the manuscript. For example: p.3 "As a case study, we propose to employ the multi-fractal network generative (MFNG) model (see Supplementary Material S6) as the underlying network model." "S6" Should be S4.

eq. 17 still uses alpha while eq. 18 uses beta.

eq. 6 the differential is still missing.

Our Response: We really appreciate the detailed proofreading of the reviewer and apologize sincerely for these previously uncaptured typos. We have made the changes accordingly.

Concerning the ability of a researcher to reproduce the work:

C-10: The authors used publicly available datasets and provide the source codes for generating the results on github. This is very nice except for the fact that there is very little explanation on how to run the code.

Our Response: We thank the reviewer for the comment. We have added more detailed instruction to run the code in github repo and Supplementary Materials (S7).

Reviewer #1 (Remarks to the Author):

My comments are mostly addressed in the last round and I accept this paper.

Reviewer #2 (Remarks to the Author):

I think the manuscript has improved dramatically and thank the authors for damping some of the claims from previous versions.

The paper is well organized and the results are indeed interesting and of possible interest to a broad set of researchers.

I note, however, that the English needs improvements in some places where sentences are constructed awkwardly.

Of more significance, I think the Figure 1 and the description of the situation in the text is not totally clear to me. First, the notation for the different possible graphs is not clear. The text refers to $G_{0'}$ but the figure shows $G_{01'}$ for the inference by network model. Which is correct?

Also, unless the nodes are not labelled, there should be an additional inferred graph by network model (with A being connected to node in the left instead of in the right) and an additional inferred graph by attack model (similar to $G_{0',2}$ but with connection to leftmost node instead of rightmost node).

Whether I am missing something or not, the figure and explanation are not clear.

Luis Amaral

Reviewer #3 (Remarks to the Author):

I am very happy with the review of the manuscript. The authors have done a tremendous job at answering my questions and clarifying my issues.

In particular, I appreciate the new introduction and discussion about the example of inference in a social network which is now much clearer.

However, I still have an issue about the new material added in the Supplementary Information: S6 Structural properties study of a real-world Russian Twitter attack by trolls in US 2016 election.

In the main text, they write "To further corroborate our assumption that degree centrality plays a key role in the attack and validate the adoption of the attack model, we have analyzed the degree centrality statistics of a real social network attack by Internet Research Agency (IRA) of Russia in Supplementary Material S6 that shows a attack model based on degree centrality can potentially serve as a reasonable abstraction of their behavior and capture the structural properties of subnetworks formed by these attackers."

My problem is that I don't understand how their analysis in the SI supports their assumption.

Indeed, the datasets they used (provided by Twitter) "include all public, nondeleted Tweets and media (e.g., images and videos) from accounts we believe are connected to state-backed information operations" (see https://about.twitter.com/en_us/values/elections-integrity.html).

This means that all users in the dataset are believed to be connected with the IRA.

However, the authors then identify only 100 accounts that are linked to the IRA (by cross-checking the list of users with an other list of IRA-backed account, if I understand correctly) who turn up to have very high degrees compared to the rest of the users.

But the comparison is not valid since all users in the original datasets are suspected to be Russian trolls.

It seems that the authors believed that the Twitter dataset contained all tweets sharing information fabricated by IRA trolls sent by all Twitter users, but, as Twitter explains on their website, the dataset contains all the tweets sent only by users believed to be IRA trolls.

It is true that when building the retweet network some users not necessarily included in the dataset will be added (people retweeted by users in the dataset, but not people having tweeted users in the dataset), but I don't think this will help the authors to support their assumption.

Unless I have made a mistake in my reasoning that the authors can explain, I would suggest to simply remove this part (I don't think there is really a need to support their assumption about the centrality of attackers anyway).

Minor remarks:

- I think the paper would be even clearer if the authors could succinctly answer the following question in the main text: "If one wants to use their method to investigate the Russian interference during the 2016 US election (for example), what kind of data would one need (from Facebook or Twitter for example) and what could their method tell us exactly?"

- In the main text, they write: "This observation is corroborated by a recent study of Russian trolls attack on Twitter[52] which found that the trolls only account for 4.9% and 6.2% of total liberal and conservative spreaders, respectively." The numbers reported in this study are about the proportion of bots, which is not necessarily the same thing than trolls, since trolls can also be real people.

We appreciate all the comments and suggestions from the reviewers. Under their guidance, we have come a long way to improve the manuscript to a much better shape. We would like to present our sincere gratitude to all reviewers and editors for their efforts.

Thank you very much for your consideration,
Yuankun Xue and Paul Bogdan

Referee 1:

My comments are mostly addressed in the last round and I accept this paper.

Our Response: We thank the reviewer for all the insightful comments, interactive discussion and all the encouragement.

Referee 2:

C-1: I think the manuscript has improved dramatically and thank the authors for damping some of the claims from previous versions. The paper is well organized and the results are indeed interesting and of possible interest to a broad set of researchers. I note, however, that the English needs improvements in some places where sentences are constructed awkwardly. Of more significance, I think the Figure 1 and the description of the situation in the text is not totally clear to me. First, the notation for the different possible graphs is not clear. The text refers to $G_{0'}$ but the figure shows $G_{0,1'}$ for the inference by network model. Which is correct? Also, unless the nodes are not labelled, there should be an additional inferred graph by network model (with A being connected to node in the left instead of in the right) and an additional inferred graph by attack model (similar to $G_{0',2}$ but with connection to leftmost node instead of rightmost node). Whether I am missing something or not, the figure and explanation are not clear.

Our Response: We are grateful to the reviewer's suggestions which guided us to significantly improve our manuscript. We thank the reviewer for helping us to capture this typo with respect to the notation $G_{0,1'}$. To make it consistent, we have changed it $G_{0,1'}$ in both main text and in the figure.

In addition, we agree with the reviewer that there can be other recovered network structures inferred by the network model. In fact, we have also mentioned this in the text:

According to the Bayesian inference principle, we infer the missing node and its links that maximize the likelihood based on the network model and the attacker's statistical behavior. By assumption (ii), the missing node inferred based on the network model will be less likely to have a higher degree. $G_{0,1'}$ therefore can be one of possible outcomes ($G_{0,2'}$ represents another possibility). Although $G_{0,1'}$ is not unique, one must choose it over many other possible configurations where node A has a higher degree. By assumption (i), the missing node inferred based on the attack can be $G_{0',1}$, $G_{0',2}$ or $G_{0',3}$ (other outcomes removed due to symmetry). However, node A is not the unique most connected node in $G_{0',1}$ and $G_{0',2}$ (i.e., only 50% chance to be chosen). Therefore, $G_{0',3}$ is the most probable outcome. Interestingly, neither $G_{0,1'}$ nor $G_{0',3}$ represents the true configuration. From the perspective of the network model, $G_{0',3}$ is a less likely structure due to the highly connected node. $G_{0,1'}$ is less likely (1/3 chance) to be the target of the attacker. Combining the knowledge of both leads us to the true G_0 in this simple case.

To make this more clear, we have regenerated the graph and added another possible inferred network to clarify our argument in the main text. We also show the modified figure here in our response.

We also thank the reviewer for the suggestion to improve English wording of our manuscript. While being limited by our written English skills as non-native speakers, we followed the suggestion and have tried our best to proofread the manuscript and improve the presentation everywhere we deem necessary.

Figure 1: **A motivating example.** An illustrative example to show the importance of combined consideration of network model and interventional behavior.

C-1: I am very happy with the review of the manuscript. The authors have done a tremendous job at answering my questions and clarifying my issues. In particular, I appreciate the new introduction and discussion about the example of inference in a social network which is now much clearer. However, I still have an issue about the new material added in the Supplementary Information: S6 Structural properties study of a real-world Russian Twitter attack by trolls in US 2016 election. In the main text, they write "To further corroborate our assumption that degree centrality plays a key role in the attack and validate the adoption of the attack model, we have analyzed the degree centrality statistics of a real social network attack by Internet Research Agency (IRA) of Russia in Supplementary Material S6 that shows a attack model based on degree centrality can potentially serve as a reasonable abstraction of their behavior and capture the structural properties of subnetworks formed by these attackers."

My problem is that I don't understand how their analysis in the SI supports their assumption. Indeed, the datasets they used (provided by Twitter) "include all public, nondeleted Tweets and media (e.g., images and videos) from accounts we believe are connected to state-backed information operations". This means that all users in the dataset are believed to be connected with the IRA. However, the authors then identify only 100 accounts that are linked to the IRA (by cross-checking the list of users with an other list of IRA-backed account, if I understand correctly) who turn up to have very high degrees compared to the rest of the users. But the comparison is not valid since all users in the original datasets are suspected to be Russian trolls. It seems that the authors believed that the Twitter dataset contained all tweets sharing information fabricated by IRA trolls sent by all Twitter users, but, as Twitter explains on their website, the dataset contains all the tweets sent only by users believed to be IRA trolls. It is true that when building the retweet network some users not necessarily included in the dataset will be added (people retweeted by users in the dataset, but not people having tweeted users in the dataset), but I don't think this will help the authors to support their assumption. Unless I have made a mistake in my reasoning that the authors can explain, I would suggest to simply remove this part (I don't think there is really a need to support their assumption about the centrality of attackers anyway).

Our Response: We are very thankful to the reviewer's comments that helped us improve significantly our manuscript throughout the entire review process. We agree with the reviewer that this retweeted network made public by Twitter is not inclusive and generated based on the identified trolls and accounts that have direct connections to them. The inclusion of this degree centrality study on this retweet network, as the reviewer insightful mentioned, is to demonstrate that trolls and bots that are injected to the social network tend to have much higher connectivity compared to regular users whose in- and out-degree as a node are mostly constrained by social exposure. However, due to the privacy and personal information protection of regular users, this dataset indeed only include trolls and Twitter accounts with immediately close ties to them. The interactions between these spreaders and regular accounts are not exposed and protected intentionally. This indeed leads to an underestimated degree centrality measurement especially for the spreaders. While searching for alternative data sources upon receipt of the review comment, we were unable to gain the access to other relevant data sources that are totally free of similar sampling bias. To avoid any scientific bias and claims based on it, we followed the reviewer's suggestion and have removed it from our Supplementary Materials. In the meantime, we will continue our search offline and perform the appropriate analysis upon the availability of expected datasets in our follow-up extension to this work. Again, we are very thankful to the reviewer's suggestion and help.

C-2: I think the paper would be even clearer If the authors could succinctly answer the following question in the main text: "If one wants to use their method to investigate the Russian inference during the 2016 US election (for example), what kind of data would one need (from Facebook or Twitter for example) and what could their method tell us exactly?"

Our Response: We are very thankful to the reviewer's comments. We added the suggested change to "Discussion". We also reproduce it here for the reviewer.

There are several key aspects that could be improved by our future work. While the assumptions made in the attack model seem plausible, the real attacks may not follow a consistent statistical pattern as the one described in Equation (2). For instance, the causal structure of the attacking sequence considered in our framework can be more sophisticated by the coordination / interaction (sharing of information) among multiple attackers co-existing in the network. Attackers may not necessarily operate under the same strategy which makes it challenging to construct consistent and accurate models to characterize their behavior. Consequently, it is important to incorporate attack strategies as part of the network inference framework (e.g., either estimating the unknown parameters of the attack model together with the network model in an EM approach or estimate them separately based on additional information when available). Since real networks can change their growth rules and possibly their self-similar structure over time (e.g., co-existence or emergent transition of small-world and multi-fractal scaling observed in complex networks [56,57], a generative model that captures all the structural features of interested networks can be difficult to build. **As a result, applying the proposed method to retrieve the latent subnetwork resulted from attacks on real-world networked systems (e.g., social network manipulation and intervention) requires time-labeled data collection. This data collection should enable reliable identification of and estimation on the statistical behavior of attackers and its variations over time (e.g., through multiple piece-wise temporal windows that correspond to different statistical modes/patterns of the attacker). Towards this end, an integration of continuous anomaly detection and data monitoring system is a must to interface with the proposed framework and other analytical tools (e.g., opinion diffusion dynamics) for identification, influence assessment and source tracking of the adversarial interventions on real-world networks.**

C-3: - In the main text, they write: "This observation is corroborated by a recent study of Russian trolls attack on Twitter[52] which found that the trolls only account for 4.9% and 6.2% of total liberal and conservative spreaders, respectively." The numbers reported in this study are about the proportion of bots, which is not necessarily the same thing than trolls, since trolls can also be real people.

Our Response: We are very thankful to the reviewer for helping us to clarify the concept. We have corrected this statement to the following:

This observation is corroborated by a recent study of Russian trolls attack on Twitter[52] which found that the injected tweet bots only account for 4.9% and 6.2% of total liberal and conservative spreaders, respectively.